

# The AFWA Dust Emissions Scheme for the GOCART Aerosol Model in WRF-Chem

Sandra L. LeGrand[1], Chris Polashenski[2,3], Theodore W. Letcher[1], Glenn A. Creighton[4], Steven E. Peckham[1], and Jeffrey D. Cetola[5]

[1]U.S. Army Engineer Research and Development Center, Hanover, NH USA
[2]Alaska Projects Office, U.S. Army Cold Regions Research and Engineering Laboratory, Fairbanks, AK USA
[3]Thayer School of Engineering, Dartmouth College, Hanover, NH USA
[4]U.S. Air Force 557th Weather Wing, 16th Weather Squadron, Offutt Air Force Base, NE USA
[5]U.S. Air Force, Joint Base Langley-Eustis, VA USA

**Correspondence:** Sandra LeGrand (Sandra.L.LeGrand@usace.army.mil)

**Abstract.** Airborne particles of mineral dust play a key role in Earth's climate system and affect human activities around the globe. The numerical weather modeling community has undertaken considerable efforts to accurately forecast these dust emissions. Here, for the first time in the literature, we thoroughly describe and document the Air Force Weather Agency (AFWA) dust emission scheme for the GOCART aerosol model within the Weather Research and Forecasting Chemistry (WRF-Chem) model and compare it to the other dust emission parameterizations available in WRF-Chem. The AFWA dust emission scheme addresses some shortcomings experienced by the earlier GOCART-WRF parameterization. Improved model physics are designed to better handle emission of fine dust particles by representing saltation bombardment. Model performance with the improved parameterization is evaluated against observations of dust emission in southwest Asia and compared to emissions predicted by the other parameterizations built into the WRF-Chem GOCART model. Results highlight the relative strengths of the available schemes, indicate the reasons for disagreement between the models, and demonstrate the need for improved soil source data.

## 1 Introduction

Airborne mineral dust particulates play a key role in Earth's radiative budget, weather and climate patterns, and biogeochemical processes (e.g., Shinn et al., 2000; Mahowald et al., 2005, 2010, 2014; DeMott et al., 2010; Ravi et al., 2011; Webb et al., 2012; Boucher et al., 2013; Huang et al., 2014; Knippertz and Stuut, 2014; Skiles et al., 2015; Wang et al., 2017a). Dust can also create hazardous air quality conditions that negatively affect health, agriculture, visibility, communication, and mobility (e.g., Goudie and Middleton, 2006; Rushing et al., 2005; McDonald and Caldwell, 2008; De Longueville et al., 2010; Okin et al., 2011; Sprigg et al., 2014; Middleton, 2017; Al-Hemoud et al., 2017). As a result, the development of accurate numerical models of dust emissions and transport is a priority for the research, operational forecasting, and hazard mitigation communities (e.g., Knippertz and Stuut, 2014; Sprigg et al., 2014; Shepherd et al., 2016).



Over the past several decades, numerous dust emission and transport models have been developed for forecasting and research purposes (e.g., Tegen and Fung, 1994; Wang et al., 2000; Woodward, 2001; Ginoux et al., 2001; Nickovic et al., 2001; In and Park, 2002; Zender, 2003; Shao, 2001; Gong, 2003; Liu et al., 2003, 2007; Tanaka and Chiba, 2005; Klose and Shao, 2012, 2013). One broadly-adopted aerosol model is The Georgia Institute of Technology-Goddard Global Ozone Chemistry

Aerosol Radiation and Transport (GOCART) model. The GOCART model includes components that represent the emission, transport, and deposition of an array of atmospheric aerosols including sea spray, combustion products, and mineral dust. In this publication, we will focus on GOCART's representation of mineral dust aerosol. More specifically, we will address one of the most important components of the mineral dust parameterization – the representation of dust emissions from the soil surface, which is the critical first step enabling their vertical movement into the atmosphere.

First, we present a brief history of relevant model development. GOCART was originally designed as a standalone, offline aerosol model driven by assimilated meteorological fields (Chin et al., 2000); however, components of the code have been added to other model frameworks since its release (e.g., Colarco et al., 2003a, b; Barnum et al., 2004; Peckham et al., 2011). In 2009, GOCART aerosol physics, including algorithms for dust emissions, transport, dry deposition, and gravitational settling, were added to the Weather Research and Forecasting Chemistry (WRF-Chem) framework. WRF-Chem is a mesoscale non-

hydrostatic Earth-system model able to simulate particulate transport and feedbacks simultaneously with the meteorological fields (Grell et al., 2005; Fast et al., 2006; Peckham et al., 2011). Many studies on WRF-Chem model performance, when configured with GOCART dust emission algorithms, have been published since this addition (e.g., Zhao et al., 2010, 2011, 2013; Bian et al., 2011; Liu et al., 2011; Kalenderski et al., 2013; Kalenderski and Stenchikov, 2016; Dipu et al., 2013; Alizadeh Choobari et al., 2013; Chen et al., 2014; Kumar et al., 2014; Jish Prakash et al., 2015; Zhang et al., 2015). Though

these studies highlight multiple useful applications of the WRF-Chem GOCART dust parameterization, many authors noted the need to tune the model for each location/event to obtain reasonable simulations of aerosol optical depth (AOD) or other dust parameters of interest. The character of the model shortcomings noted by prior studies indicated potential issues with the representation of dust availability (source strength), calculation of dust emissions as a function of wind speed, or both.

In 2011, researchers from the Air Force Weather Agency (AFWA), now designated the 557th Weather Wing, and Atmo-

spheric and Environmental Research, Inc. (AER) began to investigate the WRF-Chem GOCART source code after noting multiple unexpected simulation pattern results for dust emission in southwest Asia. Closer inspection revealed issues with the parameterization of dust emissions, which rendered the original GOCART model dust output invalid under certain environmental conditions. As a result, an alternative dust emission scheme option was developed to augment the WRF-Chem GOCART code. Several journal articles briefly discuss the use of the AER and AFWA modifications (e.g., Su and Fung, 2015; Wang et

al., 2015; Teixeira et al., 2016; Rizza et al., 2016; Fountoukis et al., 2016; Flaounas et al., 2016; Uzan et al., 2016; Nabavi, 2017; Cremades et al., 2017), but full documentation of the AFWA scheme has not yet been published. The purpose of this publication is, therefore, to document for the broader modeling community the alternate dust emission scheme (hereafter referred to as the AFWA scheme), its intended use, and how it compares with the other available dust emission schemes included in WRF-Chem GOCART. The primary objectives of this paper are threefold: 1) to discuss potential issues in simulations using

the original WRF-Chem GOCART dust emission scheme (hereafter referred to as the GOCART-WRF scheme) that motivated



development of the AFWA scheme, 2) to fully describe the algorithms comprising the AFWA scheme, and 3) to document, evaluate, and discuss the differences between dust emission simulations produced using the three available WRF-Chem dust emission schemes.

To support the objectives of this paper, we provide a full documentation of the GOCART-WRF dust emission scheme,
including changes that have been made to the code since Ginoux et al. (2001) and Ginoux et al. (2004) that are otherwise incompletely documented in the literature. Next, we detail shortcomings with the original GOCART dust emission scheme (even as revised) and discuss how the AFWA scheme attempts to address these issues, including full documentation of the AFWA dust emission scheme. For completeness, we also discuss the third dust emission scheme currently available for WRF-Chem GOCART, commonly referred to as the University of Cologne (UoC) emission scheme (based on Shao, 2001, 2004; Shao
et al., 2011) and how it might be expected to perform differently by comparing its parameterization with the AFWA scheme. We then present a case study WRF-Chem simulation of dust emissions from southwest Asia for a dust event that occurred during January 2010. We use this case study to illustrate the performance of the three dust options included in all releases of WRF-Chem since version 3.6.1, and follow with a discussion of the possible reasons for the discrepancies between the model outputs. We conclude with a recommendation that future model development focus on improving the soil characterization
datasets that form the foundation of both the AFWA and UoC schemes.

The paper is organized as follows: In section 2, a brief background on the physics of dust emission is provided. In section 3, the three dust emission schemes included in the WRF-Chem model are described. In sections 4 and 5, the model configuration and data analysis methods are described. In sections 6 and 7, the results of the study are presented and discussed. Conclusions are presented in section 8.

## 20   2   Background: The physics of the emission of dust

Soil particles mobilize when lift, drag, and impact forces overcome the gravitational and inter-particle cohesive forces holding them to the soil bed (e.g., Bagnold, 1941; Kok et al., 2012, and references within). The forces that lead to dust emission can be thought of in terms of three processes, (1) aerodynamic lift, (2) saltation bombardment, and (3) particle disaggregation. Aerodynamic lift (1) is the process by which wind shear forces directly act upon dust particles at the surface. When lift and
drag forces overcome gravitational and cohesive forces, mobilization results. Because inter-particle cohesive forces on particles smaller than 60-70 μm are generally much larger than aerodynamic forces, dust-sized (∼0.1-10 μm) particles are rarely lofted directly by the wind (Chepil, 1945; Gillette and Passi, 1988; Shao, 2001). Instead, aerodynamic lift is most efficient at lofting slightly larger particles. Fine sand grains or aggregates on the order of 60 to 70 μm are the first to detach as wind speeds increase. Direct mobilization of these larger, sand-sized particles brings about dust-sized particle mobilization through the other modes
– saltation bombardment and particle disaggregation. Once lofted, the larger sand-sized particles undergo saltation; a process in which mobilized particles too heavy to remain in suspension fall back upon the land surface with ballistic trajectories, after being accelerated by the airstream. The impact energy from the collisions can engage new particles into saltation, creating a positive feedback. Dust emission by saltation bombardment (2) occurs in this latter case, when the impact energy from a





previously mobilized particle striking the soil surface imparts sufficient force to overcome the cohesive and gravitational forces binding particles to the surface (Gillette, 1981; Alfaro et al., 1997). Saltation bombardment is the most common mode for mobilization of smaller dust-sized particles because bombardment can effectively transfer wind energy to break bonds among particles too strongly cohered to mobilize by direct wind shear forcing (aerodynamic lift). Modeling saltation bombardment

can be challenging because it requires correctly modeling both wind shear mobilization of larger particles and bombardment interactions between particles of differing size. The third process, particle disaggregation (3) is mechanistically similar to saltation bombardment. Again, the initial mobilization of large particles is due to wind shear forces, and emission of dust-sized particles is caused by energy dissipation during collisions. Instead of collisions mobilizing dust particles from the soil surface, however, the dust emitted is part of the saltating particle and may originate from dust coatings on solid particles or

clay aggregates disintegrating during collisions (e.g., Chappell et al., 2008; Bullard et al., 2007). Saltation impacts in this case break apart the binding of mobilized soil aggregates and eject finer dust-sized particles into the air. The disaggregation mode can be a significant source of aerosol particles under select soil conditions and is challenging to effectively model without a priori knowledge of soil conditions. To adequately represent dust production processes, an emission scheme must account in some way for (at least) the second and third emission modes (saltation and disaggregation). Doing so requires representing the

mobilization of saltating grains through wind shear (the first emission mode), the transfer of energy from saltating grains to dust particle ejection during collisions, and the resistance of the soil to sandblasting during these energetic collisions.

## 3   Model description: The dust emission modeling schemes in WRF-Chem GOCART

At present there are three different dust emission schemes built into the WRF-Chem model, the original GOCART-WRF scheme (*dust_opt=1*), the AFWA scheme (*dust_opt=3*), and the University of Cologne (UoC) scheme (*dust_opt=4*). The

*dust_opt=2* setting is not applicable to GOCART and has since been disabled. As of this writing, there are 17 baseline versions of WRF-Chem available to the public (starting with version 3.2). The GOCART-WRF scheme is available in all versions, the AFWA scheme was released in version 3.4, and the UoC scheme was released in version 3.6.1. Various changes have been made to each of the dust emission schemes over time. Both the changes, and the original nature of the schemes have been incompletely documented in the literature. The primary purpose of this publication is to document the AFWA scheme.

However, an attempt is made to identify and highlight portions of the other schemes that are undocumented or are implemented inconsistently with existing documentation.

### 3.1   The GOCART-WRF dust emission scheme

#### 3.1.1   The original, standalone GOCART dust emission scheme

The version of the dust emission scheme originally described by Ginoux et al. (2001) is referred to here as the "original" dust

emission scheme, for lack of a better term in common usage. The scheme was incorporated into the standalone GOCART model, and, in later versions, embedded in WRF-Chem version 3.2. In WRF-Chem it is called by setting *dust_opt=1* in the





namelist configuration file. We refer to the model after its incorporation into WRF-Chem as the GOCART-WRF scheme. This section refers to the model in general, while the next section (3.1.2) refers to the GOCART-WRF version specifically.

The original GOCART dust emission scheme is popular with the broader modeling community because it does not require difficult-to-obtain soil or surface characteristics to run (e.g., soil composition, micro- or macro-scale terrain roughness, vegeta-
tion type and spacing, soil aggregate strength, etc.). Instead, geographic variability in substrate erodibility is fixed by a simple, topographically-based, internally-calculated source function. Erodible soil makeup is then fixed to a constant mix of sand, silt, and clay. Wind speed, soil moisture, air density, and generalized soil traits are the only necessary inputs for its dust emission flux calculation, and these are determined from variables readily available in most numerical weather models. This standalone nature of the original GOCART dust model has made it an attractive choice for research and operational centers in need of
regional- or global-scale dust products (e.g., Barnum et al., 2004; Colarco et al., 2010; Lu et al., 2013; Peters-Lidard et al., 2015).

We first summarize the original GOCART dust emission scheme as it was documented by Ginoux et al. (2001). The original GOCART dust emission scheme calculates dust particle emissions separately for discrete bins of soil grain sizes (referred to as size bins), based on wind speed and soil moisture. Emissions are calculated using an equation modified from work originally
by Tegen and Fung (1994), and with basis in Gillette and Passi (1988). The scheme is empirical, since its equations represent a direct conversion from wind speed to dust emission, rather than using wind speed to calculate a saltating particle flux and then using the saltating particle flux to determine dust emissions, as the physics of dust emission by saltation bombardment discussed in section 2.1 would motivate. The impacts of saltation bombardment processes on mobilization are not necessarily omitted – rather they are internalized in the relationship between wind speed and emissions. Physically, this simplification is
akin to fixing the balance between the modes of emission to be constant for all locations. In the original Ginoux et al. (2001) description, seven size bins, representing soil grains with effective particle diameters ($D_p$) of 0.1 to 6 μm (i.e., clay and small silt-sized particles) were used to represent aerosol sizes most important on a global scale. No size bins were tracked to account for mobilization of saltation particle sizes (e.g., $D_p > 10$ μm). Emission flux values for each size bin ($F_p$; kg m$^{-2}$ s$^{-1}$) were obtained using

$$F_p = \begin{cases} C S s_p U^2 \left( U - U\left(D_p, \theta_s\right) \right), & U > U_t\left(D_p, \theta_s\right) \\ 0, & U \leq U_t\left(D_p, \theta_s\right) \end{cases} \qquad (1)$$

where $C$ is a dimensional proportionality constant (default set to 10$^{-6}$ g s$^2$ m$^{-5}$ in Ginoux et al. (2001); note that units of kg s$^2$ m$^{-5}$ in the WRF-Chem model change the value to order 10$^{-9}$), $S$ is a unitless dust source strength function indicating availability of entrain-able particles, $s_p$ is the mass fraction of emittable dust from the soil separate class (i.e., sand, silt, or clay) of size group $p$ at the soil surface, $U$ is the horizontal wind speed at 10 m, and $U_t\left(D_p, \theta_s\right)$ is the threshold 10m wind velocity required
for initiating erosion.





The threshold wind velocity $U_t(D_p, \theta_s)$ is first derived for dry soil conditions based on particle diameter, $D_p$, and then adjusted for soil surface wetness in terms of degree of saturation, $\theta_s$. In the original scheme, threshold wind velocity for dry soil, $U_t(D_p)$, was determined by

$$U_t(D_p) = A\sqrt{\frac{\rho_p - \rho_a}{\rho_a}gD_p} \tag{2}$$

where $A = 6.5$ is a dimensionless tuning parameter, $D_p$ is the particle diameter, $g$ is gravitational acceleration, and $\rho_p$, $\rho_a$ are the particle and air density, respectively. As we will note momentarily, this realization of the $U_t(D_p)$ function was changed prior to the incorporation of the original GOCART scheme into WRF-Chem in version 3.2. A conditional statement was used to correct the threshold wind velocity for soil moisture. No erosion occurs if the soil surface wetness is above 0.5. If it is below 0.5, $U_t(D_p)$ is corrected for soil moisture following

$$U_t(D_p, \theta_s) = \begin{cases} U_t(D_p) \times (1.2 + 0.2\log_{10}\theta_s), & \theta_s < 0.5 \\ \infty, & \theta_s \geq 0.5 \end{cases} \tag{3}$$

Curiously, this means that the value of the correction factor varies from 0 to 1.2, equaling 1 at a soil moisture content of 10%. This effectively treats the threshold velocity for dry soil, calculated in Eq. (3), as if it were for soil having a moisture content of 10% and could result in adjusted threshold velocities that are actually below the dry soil calculated velocity for very low soil moisture conditions. The impact is, however, minimized since soil moisture is typically restricted from falling below the hygroscopic point in most numerical weather models, which prevents extremely low soil moisture values from being reached.

$S$, the unitless dust source strength function used in the calculation of $F_p$ in Eq. (1), was added as a stand in for difficult-to-obtain soil surface characteristics necessary for describing availability of loose erodible soil material. $S$ was determined based on the degree of topographic relief surrounding a model cell, based on the premise that dust material is often generated in alluvial processes and accumulates in low points, according to

$$S = \left(\frac{z_{max} - z_i}{z_{max} - z_{min}}\right)^5 \tag{4}$$

where $z_i$ is the elevation of the cell and $z_{max}$ and $z_{min}$ are the maximum and minimum elevation in the surrounding 10° x 10° area, respectively. $S$ is set to zero anywhere bare soil is not indicated by AVHRR data (Defries and Townshend, 1999).

Dust mass flux values $F_p$ calculated from the scheme are used to represent dust mass flux injected into the lowest atmospheric model level. Separate schemes for atmospheric transport and removal from the atmosphere are used to estimate mass concentrations of dust aloft in the atmosphere.

### 3.1.2 The GOCART-WRF dust emission scheme and its updates

The GOCART-WRF dust emission scheme was first incorporated into WRF-Chem version 3.2. and is called by setting *dust_opt=1*. Although the GOCART-WRF emission scheme is based on the original GOCART scheme described in Ginoux





et al. (2001), the version embedded in WRF-Chem (from version 3.2 through the current release version 4.0) contains some important modifications from the original Ginoux et al. (2001) descriptions summarized above. The scheme has been updated several times since its introduction into WRF-Chem version 3.2, and these changes are incompletely documented in the literature. The most notable modification is a change in the threshold wind velocity equation for dry soil (Eq. (2)), which is used

after being adjusted for soil moisture (Eq. (3)) in Eq. (1) to calculate particle emission flux. This change was made prior to the incorporation of the GOCART model into WRF-Chem version 3.2, and is therefore present in all versions of WRF-Chem that include the GOCART-WRF dust emission scheme. We discuss the replacement (Eq. (5)) in detail starting in the next paragraph. In reviewing the source code, we also noted other changes to the GOCART-WRF dust emission scheme relative to the description in Ginoux et al. (2001), which we document for the community as follows:

1. A change in the number of dust emission size bins (now 5) and the size range for those bins (now 0.1-20 μm) from 7 bins ranging 0.1 to 6 μm described in Ginoux et al. (2001). This change was made prior to incorporation into WRF-Chem. All versions of GOCART-WRF use the 5 bins.

   2. Use of a precalculated source strength function $S$ (stored in the code as the variable *EROD*), which is read in and interpolated to the model grid by the WRF-Chem preprocessor. The developers who did the initial code implementation

provided static *EROD* values calculated using Eq. (4), a 1-degree resolution elevation dataset, and the AVHRR-based vegetation mask. This dataset was later replaced by an alternate version derived from quarter-degree resolution elevation data in April 2012 (change coincided with the community release of WRF-Chem version 3.4).

   3. A simplification of soil makeup is incorporated into the dust emission flux (Eq. (1)). All alluvium available for lofting is assumed to have a constant distribution of 50% sand, 25% silt, and 25% clay. The *EROD* parameter provided by the

WRF-Chem preprocessor is stored as a 2-layer variable, with the first layer equal to $0.5S$ and the second layer equal to $0.25S$. Each dust size bin is assigned a indicator value (*ipoint* in the code) to signify whether the bin represents clay-, silt-, or sand-sized grains. Layer 1 is used to parameterize the $S$ term in Eq. (1) for size bins that fall into the sand-sized category, and Layer 2 is used for the clay and silt categories. The effect is that the net $S$ value never exceeds 0.50 because none of the default dust size bins represent sand-sized particles, and the sand fraction is dictated to comprise half the

erodible soil mass.

   4. The addition and later removal of a tuning constant which multiplies the emitted dust mass by 0.2 as it is being added to the first atmospheric model layer. This tuning constant may produce unexpected results because it does not alter the dust emission flux values output to the WRF-Chem history file, even as it substantially reduces dust entrained into the atmosphere. The tuning constant is present in versions 3.3 through 3.8, but is not present in versions 3.2, 3.2.1, and

3.8.1–4.0.

   5. The dimensional proportionality constant, $C$, present in Eq. (1) here and referenced in Eq. (2) of Ginoux et al. (2001) (which is often treated as a tuning constant by users) is prescribed as $0.8 \times 10^{-9}$ kg s$^{-2}$ m$^{-5}$, slightly different from the value of $1.0 \times 10^{-9}$ kg s$^{-2}$ m$^{-5}$ provided in Ginoux et al. (2001).





6. Soil moisture values passed in by the WRF-Chem framework are converted from volumetric water content ($\theta_v$) to degree of saturation ($\theta_s$) for use in Eq. (3) via $\theta_s = \theta_v/\phi$, where $\phi$ is the porosity of the soil medium.

7. The threshold soil moisture value used to restrict dust lofting in Eq. (3) was set to 0.2 in WRF-Chem versions 3.2–3.4.1 but later changed to 0.5, bringing the value into agreement with Ginoux et al. (2001) in versions 3.5–4.0.

8. The most substantive change in the GOCART-WRF dust emission scheme relative to the description in Ginoux et al. (2001), however, is a revision to how the threshold wind speed required for dust emissions is calculated. In the original model description, the parameter was calculated according to Eqs. (2) and (3), above. Note that at a given soil moisture content, threshold wind velocity in this formulation is always greater for larger particle diameters. Publications quickly noted that such a parameterization did not empirically reflect known behavior (e.g., Colarco et al., 2003a). Well-
established experimental observations instead show particles below $\sim 60\ \mu m$ in size exhibit higher threshold wind speeds with decreasing diameter due to the increasingly dominant influence of cohesive effects on smaller particle binding (e.g., Alfaro et al., 1998). The modified version, which has been in GOCART-WRF since it was first incorporated into WRF-Chem version 3.2 and later, replaced this method for calculating the threshold wind speed, $U_t$, with an equation from Marticorena and Bergametti (1995, MB95 hereafter), which was derived in terms of friction velocity, $u_*$, instead of 10m
wind speed:

$$u_{*t}\left(D_p\right) = 0.129 \frac{\left(\frac{\rho_p g D_p}{\rho_a}\right)^{0.5}\left(1 + \frac{0.006}{\rho_p g D_p^{2.5}}\right)^{0.5}}{\left[1.928\left(a\left(D_p\right)^x + b\right)^{0.092} - 1\right]^{0.5}} \tag{5}$$

where $D_p$ is the particle diameter in bin $p$, $g$ is acceleration due to gravity, $\rho_p$ is the particle density in bin $p$, $\rho_a$ is air density, $x = 1.56$, $a = 1331$ cm$^{-x}$, and $b = 0.38$. (Note that in the model implementation, the coefficient 0.129 is represented as 0.0013 due to rounding and due to the fact that particle diameters are initially ingested by the scheme in
units of m for consistency with other aerosol parameters handled by the WRF-Chem framework. The rounding has no material impact on the output).

The switch to this revised scheme improved the model's ability to reproduce the known behavior of small diameter particles – specifically by requiring higher threshold wind speeds for fine particle mobilization. The revision, therefore, produced empirically improved results. From a physical standpoint, however, motivation for the use of the MB95
equation is strained (Colarco et al., 2003a). The MB95 equation was designed to determine the threshold for initiating wind shear-based saltation of grains – not to represent the threshold for wind shear-based emission of finer-grained dust particles from the surface. This, as we discussed previously, is primarily caused by saltation bombardment and particle disaggregation.

The change from Ginoux et al. (2001) to MB95 methods for deriving threshold speed also resulted in what may have been
an inadvertent shift from a calculation of threshold speed in terms of standard 10m wind speed ($U_t$) to one in terms of friction



velocity ($u_{*t}$). Although $U_t$ and $u_{*t}$ are both expressed in terms of speed, values of $U$ are typically an order of magnitude, or more, greater than their equivalent $u_*$. The revised GOCART-WRF scheme did not incorporate equations to convert resultant $u_*$ thresholds to equivalent horizontal wind speeds, an issue noted in an earlier implementation of the model (Colarco et al., 2003a). Since Eq. (1) is two part, dependent on the relationship between threshold speed and current wind speed, the

substitution of $u_* t$ where $U_t$ had formerly been used results in emissions not being set to zero until wind speeds are below a very low threshold magnitude speed (the threshold expressed in terms of friction velocity). The result is spurious lofting of dust at low wind speeds. The substitution of $u_{*t}$ where $U_t$ had formerly been used also alters modeled emissions above the threshold speed. This occurs because the $U_t(D_p, \theta_s)$ parameter in the GOCART-WRF dust emission scheme, represented in Eq. (1), is effectively absent (i.e., has near zero value) for larger speeds when it is determined using a threshold in terms of

friction velocity ($u_*$), as is computed from MB95. Simulated dust emission rates using the revised scheme are then effectively proportional to the cube of the wind speed over areas with dust source regions (i.e., $S > 0$ as defined in Eq. (2)). A relationship of this character cannot match observed behavior over wide ranges in wind speed but could be tuned to match emissions under narrow sets of conditions.

Modifying $U_t(D_p, \theta_s)$ to convert from friction velocity to near surface wind speed in the dust emissions flux equation,

however, is unlikely to fully resolve observed issues. The character of the emission flux – which is dependent on an empirically motivated, but physically invalid use of the MB95 equation – can likely be better represented. The logical next step in model improvements would be to continue to use the MB95 equation, but to use it in a more physically realistic manner; to represent a saltation flux threshold. The saltation flux could then be calculated, and a new parameterization could be used to convert between saltation flux and emissions from bombardment and/or disaggregation processes. Such a parameterization

would demand the addition of more particle size bins for handling the saltating particles since particle sizes represented in the GOCART-WRF emission scheme are only representative of emitted dust particle sizes. This is the approach taken by the AFWA scheme described in this paper.

## 3.2    The Air Force Weather Agency (AFWA) dust emission scheme

The AFWA scheme is a modified version of the MB95 saltation-based dust emission scheme which handles dust emission as a

two-part process, wherein large particle saltation is triggered by wind shear and leads to fine particle emission by bombardment and disaggregation. The equations for the AFWA scheme are derived in terms of friction velocity, $u_*$, and include the static threshold friction velocity required for particle entrainment ($u_{*t}$), the horizontal saltation flux, the resultant bulk vertical dust flux, the emitted dust particle size distribution, and the size resolved emitted dust flux. Similar to the GOCART-WRF scheme, particles are divided into a predetermined number of bins based on their effective aerosol size. The AFWA scheme, how-

ever, utilizes an independent series of bins for saltation-based processes and emitted dust, allowing dust emission by saltation bombardment and particle disaggregation to be better represented (and saving the resources that would have been required to compute advection of saltation particles, which are too large for significant long distance advection). Attributes associated with the ten saltation size bins and five dust size bins are given in Tables 1 and 2, respectively. Dust particle densities and effective diameters are consistent with those used in the GOCART-WRF configuration. We maintain the assumption that all clay soil





particles have a density of 2.5 g cm$^{-3}$, and that all non-clay soil particles have a density of 2.65 g cm$^{-3}$, the particle density of quartz. Lastly, the effective diameters used in the following equations are assumed to be in units of cm and are denoted as $D_{s,p}$ and $D_{d,p}$ for the saltation and dust size bins, respectively.

Saltation processes for a given size bin initiate and cease during the simulation as $u_*$ exceeds or falls below sized-resolved values of $u_{*t}$, respectively. Semi-empirical values for $u_{*t}$ (in units of cm s$^{-1}$) are calculated according to the expression of MB95, which is identical to the equation used in the GOCART-WRF scheme above (Eq. (5)) and repeated here for readers' convenience,

$$u_{*t}(D_{s,p}) = 0.129 \frac{\left(\frac{\rho_{s,p}gD_{s,p}}{\rho_a}\right)^{0.5}\left(1 + \frac{0.006}{\rho_{s,p}gD_{s,p}^{2.5}}\right)^{0.5}}{\left[1.928\left(a\left(D_{s,p}\right)^x + b\right)^{0.092} - 1\right]^{0.5}} \qquad \text{(5, repeated)}$$

where $g$ is acceleration due to gravity, $\rho_{s,p}$ is the particle density of the saltation size bin $s$, $\rho_a$ is air density, $x = 1.56$, $a = 1331$ cm$^{-x}$, and $b = 0.38$. We note that this is exactly the same equation that is used in the revised version of the GOCART-WRF scheme above, only here it is used to produce values that will be treated as friction velocities, as intended. As before, note that in the model implementation, the coefficient 0.129 is represented as 0.0013, due to rounding and particle diameter unit conversion from m to cm. Similar to the GOCART-WRF scheme, a correction function, $f(\theta)$, is applied to the threshold friction velocity to account for the effects of soil moisture on particle cohesion. The equation used for the AFWA scheme is different from that used in the GOCART-WRF scheme and was originally described by Fécan et al. (1999),

$$u_{*t,s,p} = u_{*t}(D_{s,p}) f(\theta) \qquad (6)$$

where

$$f(\theta) = \begin{cases} \sqrt{1 + 1.21\left(\theta_g - \theta_g'\right)^{0.68}}, & \theta_g > \theta_g' \\ 1, & \theta_g \leq \theta_g' \end{cases} \qquad (7)$$

$\theta_g$ is the gravimetric soil moisture fraction, and $\theta_g'$ is the fraction of soil moisture able to be absorbed before capillary forces begin to markedly influence particle detachment. As per Fécan et al. (1999), we assume,

$$\theta_g' = 0.0014(100c_s)^2 + 0.17(100c_s) \qquad (8)$$

where $c_s$ is the soil clay content mass fraction determined from soil particle size information for the surface layer of soil (0–30cm), originally derived from the global Food and Agriculture Organization (FAO) digital Soil Map of the World (SMW) by Reynolds et al. (2000), available at the NASA Land Data Assimilation System (LDAS) https://ldas.gsfc.nasa.gov/gldas/GLDASsoils.php. The original 5-minute grid of this data product is interpolated to a 1km grid for use in this application.





In order to provide the gravimetric water content ($\theta_g$) terms demanded in Eqs. (6)–(8), volumetric water content ($\theta_v$) soil moisture values provided by WRF-Chem are converted through the following relationship

$$\theta_g = \frac{\theta_v \rho_w}{(2.65 - 0.15c_s)(1 - \phi)} \tag{9}$$

where $\rho_w$ is water density equal to 1.0 g cm$^{-3}$, $\phi$ is the porosity of the soil medium, and the $2.65 - 0.15c_s$ term represents the

soil density.

Once time varying $u_{*t,s,p}$ values are known, the momentum transfer effects of wind shear and saltating grain impact shear on simulated dust emission are accounted for across varying wind speeds greater than the threshold speed via a horizontal saltation flux equation. The saltation flux is then used to calculate dust emission. First, particle size-dependent saltation fluxes ($H(D_{s,p})$; g cm$^{-1}$ s$^{-1}$) are calculated by,

$$H(D_{s,p}) = \begin{cases} C_{mb}\frac{\rho_a}{g}u_*^3\left(1 + \frac{u_{*t,s,p}}{u_*}\right)\left(1 - \frac{u_{*t,s,p}^2}{u_*^2}\right), & u_* > u_{*t,s,p} \\ 0, & u_* \le u_{*t,s,p} \end{cases} \tag{10}$$

where $C_{mb}$ is an empirical proportionality constant set to 1.0. Of note, the original MB95 study utilized a proportionality constant of 2.61 in accordance with findings by White (1979). In the model implementation, we have adopted $C_{mb} = 1.0$ as suggested by Marticorena et al. (1997) and Darmenova et al. (2009) based on more extensive wind tunnel measurements. The $H(D_{s,p})$ values are then integrated over particle sizes to obtain the total streamwise horizontal saltation flux ($G$).

Estimated contributions of each saltation size bin to total saltation flux ($G$) depend upon the surficial coverage of particles in each saltation particle size bin as a fraction of the total surface area of the soil bed. As with common land surface modeling practices (e.g., Mitchell, 2005; Wang et al., 2017b), the WRF-Chem land surface model assumes that all particles comprising the soil column belong to one of three U.S. Department of Agriculture (USDA) defined soil separate categories based on particle size: sand (50 to 2000 μm), silt (2 to 50 μm), or clay ($\le 2$ μm). Instead of the fixed soil separate fractions used in the

GOCART-WRF scheme, the makeup of soil in the AFWA model is set using the soil particle size information for the surface layer of soil (0–30cm) originally derived from the global FAO digital Soil Map of the World (SMW) by Reynolds et al. (2000). Again, the original 5-minute grid of this data product is interpolated to a 1km grid for use in this application. Starting from mass fractions in the sand, silt, and clay soil categories, we diagnose relative weighting factors for each size bin ($dS_{rel}(D_{s,p})$). The mass fractions are further distributed amongst the saltation size bins following the approach of Tegen and Fung (1994).

Linear mass distributions are assumed for the sand and silt categories while a lognormal mass distribution is assumed for clay. Size-resolved basal surface coverage fractions ($dS_{SFC}(D_{s,p})$) are then diagnosed from the mass distribution of particles in the surface soil ($dM(D_{s,p})$) as follows,

$$dS_{SFC}(D_{s,p}) = \frac{dM(D_{s,p})}{\frac{2}{3}\rho_{s,p}D_{s,p}} \tag{11}$$





Bin specific values of $\mathrm{d}M\left(D_{s,p}\right)$ are set by multiplying the bin specific mass fraction of a size bin's corresponding soil separate class ($s_{frac}$; Table 1) by the mass fraction of the matching soil separate category at each domain grid-point.

Saltation bin-specific weighting factors are then found by taking the ratio of $\mathrm{d}S_{SFC}\left(D_{s,p}\right)$ to the total basal surface area of the soil bed ($N_{SFC}$),

$$\mathrm{d}S_{rel}\left(D_{s,p}\right) = \frac{\mathrm{d}S_{SFC}\left(D_{s,p}\right)}{N_{SFC}} \tag{12}$$

where

$$N_{SFC} = \Sigma_{s,p}\left[\mathrm{d}S_{SFC}\left(D_{s,p}\right)\right]$$

The total streamwise horizontal saltation flux is then computed via,

$$G = \Sigma_{s,p}\left[H\left(D_{s,p}\right)\mathrm{d}S_{rel}\left(D_{s,p}\right)\right] \tag{13}$$

To estimate the bulk emission flux of dust ($F_B$; g cm$^{-2}$ s$^{-1}$) triggered by saltation, the AFWA scheme utilizes both the dust source strength parameterization ($S$; *EROD* in the code) from the GOCART-WRF function (Eq. (4)) and a reformatted version of the sandblasting efficiency approach from MB95. Because the source strength function provided by the WRF-Chem preprocesser is stored as a 2-layered variable (a simplification specific to GOCART-WRF), the source strength term is set in the AFWA scheme simply by multiplying the the second layer of the *EROD* parameter (equal to $(0.25S)$ by 4, resulting in a source term varying from 0–1,

$$F_B = GS\beta \tag{14}$$

where the sandblasting efficiency ($\beta$) is given by $\beta = 10^{0.134(c_s)-6}$ and has units of cm$^{-1}$. As before, $c_s$ is the soil clay content mass fraction determined from the FAO-SMW data. We note that the impact of the soil in the model is small, since the factor $\beta$ varies from only $1.00\mathrm{x}10^{-6}$ cm$^{-1}$ to $1.08\mathrm{x}10^{-6}$ cm$^{-1}$ over clay fraction of 0–0.2, and that this may underrepresent the importance of the soil type. Even considering the full theoretical range of clay fraction of 0–1, which is rare over large domains in practice, the factor $\beta$ only ranges from $1.00\mathrm{x}10^{-6}$ cm$^{-1}$ to $1.36\mathrm{x}10^{-6}$ cm$^{-1}$.

Once total dust emission ($F_B$) is determined, emissions are distributed amongst suspended dust size bins using the Kok et al. (2011) brittle fragmentation theory. Following the Kok et al. (2011) technique, we assume impacted soil aggregates will fracture in a manner similar to glass or gypsum material. Suspended dust distribution weighting factors ($\kappa_{d,p}$) are diagnosed by taking the ratio of the normalized volume distributions of each dust size bin ($\mathrm{d}V_{d,p}$) to the total normalized volume distribution of emitted dust ($N_V$),

$$\kappa_{d,p} = \frac{\mathrm{d}V_{d,p}}{N_V} \tag{15}$$





where

$$\mathrm{d}V_{d,p} = \frac{D_{d,p}}{c_v} \left[ 1 + \mathrm{erf}\left( \frac{\ln\left(D_{d,p}/\bar{D}_m\right)}{\sqrt{2}\ln\sigma_s} \right) \right] \exp\left[ -\left( \frac{D_{d,p}}{\lambda} \right)^3 \right] \ln\frac{D_{d,p\_max}}{D_{d,p\_min}}$$

$$N_V = \Sigma_{d,p}\left[\mathrm{d}V_{d,p}\right]$$

$\bar{D}_m$ is the dust particle mass median diameter equal to $3.4 \times 10^{-4}$ cm, $\sigma_s$ is the geometric standard deviation equal to 3.0, $c_v$
is a normalization constant equal to $12.62 \times 10^{-4}$ cm, $\lambda$ is the crack propagation length equal to $12.0 \times 10^{-4}$, erf is the error
function, and $D_{d,p\_max}$ and $D_{d,p\_min}$ are the maximum and minimum effective diameters represented by the dust size bin,
respectively. Resultant values for Eq. (15) are currently prescribed in the AFWA scheme since not all FORTRAN compilers
are able to process the error function. The code, however, is still present (commented out) should a user wish to change the
default dust size bin ranges. Finally, size-resolved dust emission fluxes ($F_{d,p}$; g cm$^{-2}$ s$^{-1}$) are obtained according to

$$F_{d,p} = F_B \kappa_{d,p} \tag{16}$$

As with the GOCART-WRF scheme, the emitted dust particles are released into the lowest atmospheric model level for
dispersion according to their respective size bins.

Four optional tuning parameters, three alternate input dataset channels, and an optional modification to the $f(\theta)$ calcu-
lation have been added to the AFWA scheme since its original debut in the WRF-Chem baseline. Table 3 provides a brief
overview of these additions, which can be set or activated through the WRF-Chem run-time configuration file (referred to as
the namelist.input file in the WRF-Chem framework), if desired. It should be noted, however, that the developers primarily
added these options to facilitate perturbations when using the scheme in a multi-model ensemble mode. Rigorous testing for
optimal tuning recommendations are beyond the scope of this paper, and the case study demonstrations provided in this report
do not make use of these optional settings (i.e., all optional tuning parameters are set to 1.0). Figure 1 presents a schematic
summary overview of the AFWA scheme, including the five major components, their required input parameters, and the con-
figurable run-time options.

An error in the number and distribution of saltation size bins was made during the implementation of the AFWA scheme
code into the WRF-Chem baseline. Current and legacy versions of the AFWA scheme (WRF-Chem versions 3.4 – 4.0) assume
nine saltation size bins, including one clay-, five silt-, and three sand-sized bins. Attributes of these alternate saltation size bins,
as implemented, are provided in Table 4. Bins 7–9 are sand-sized bins with effective diameters of 69, 131, and 250 μm, respec-
tively. These same bins are also configured so their combined mass fraction constitutes 100% of the possible sand mass fraction
distribution. This particular setting implies the sand portion of the soil surface is entirely composed of fine sands, and increases
the strength of the saltation bin-specific weighting factors (Eq. (11)) for these bins. Future releases of the WRF-Chem AFWA
code will be corrected for this discrepancy; however, users can amend the saltation bin configuration in their existing code by





altering the settings for the *ngsalt*, *reff_salt*, *den_salt*, *spoint*, and *frac_salt* parameters in the *module_data_gocart_dust.F* file according to Table 1.

### 3.3 The University of Cologne (UoC) dust emission scheme

WRF-Chem's third standard dust emission model, commonly referred to as the University of Cologne (UoC) model, is activated
by using *dust_opt=4* in the WRF-Chem namelist. The UoC model is documented in Shao (2001) and later papers by the same author (Shao (2004); Shao et al. (2011)) that describe sub-option sets of varying complexity. These sub-options are activated by setting the value of the variable *dust_scheme* in the namelist.input file. We will note these sub-options and the references describing them here as S01, S04, and S11, respectively, in order from most complex to most simplified parameterization of dust emission processes. Here we describe key aspects of the implementation of the UoC model and make comparisons with the
AFWA scheme. The comparison primes us for understanding the differences between the model outputs discussed in Sections 4–6.

The UoC model follows the same general approach as the AFWA model. Both schemes simulate the physics of dust emission by first calculating a threshold friction velocity for particle saltation, then using that threshold friction velocity to determine saltation flux, and finally calculating emissions of dust particles caused by saltation processes (e.g., bombardment). Both
schemes also use the same size-resolved dust emission bins. The more sophisticated UoC schemes also use size-resolved saltation particle bins to evaluate dust emission from saltating particles of different sizes.

The calculation of the threshold friction velocity for initiation of particle saltation used by the UoC schemes is of significantly different form, compared to that used in the AFWA scheme, but has similar output in terms of calculated threshold friction velocity ($u_{*t}$) under a given set of forcing conditions. Equation (5) and Eq. (17) serve this equivalent function for the AFWA
and UoC schemes, respectively, with

$$u_{*t}(d) = \sqrt{A_N \left( \sigma_p g d + \frac{\gamma_c}{\rho_p d} \right)} \qquad (17)$$

in the UoC scheme, where $\sigma_p$ = the ratio of particle to air density, $g$ is the gravitational constant, $d$ is particle diameter, $\rho_p$ is the particle density, and the $A_N = 0.0123$ and $\gamma_c = 1.65 \times 10^{-4}$ kg s$^{-2}$ are constant. Equation (17) here is replicated from Eq. (24) in Shao and Lu (2000), as referenced by S01 and S11. As we will note in documenting code discrepancies below, $\gamma_c$
is set to $1.65 \times 10^{-4}$ kg s$^{-2}$ in the code, while it is specified as $3.0 \times 10^{-4}$ kg s$^{-2}$ in Shao and Lu (2000). Note that here $d$ is particle diameter, as opposed to $D_p$ above. We have chosen to make this change to preserve the variable name choices in the UoC papers (S01, S04, S11) here while discussing the UoC schemes, which results in some factors being represented by two variables within this paper. Please see the variable list in appendix A for a complete listing of variable names, as well as the schemes and equations in which they apply.

After establishing the dry soil threshold friction velocity ($u_{*t}(d)$), all versions of the UoC model correct for the influence of soil moisture on threshold friction velocity using the parameterization in Fécan et al. (1999). This soil moisture correction is similar to the approach taken in the AFWA scheme (see Eqs. (6)–(9)). Unlike the AFWA approach, however, the UoC scheme





maintains soil moisture in terms of the volumetric soil moisture ($\theta_v$) and varies the empirical constants of Eq. (7) as a function of soil texture. In the UoC model, an additional correction factor, titled the roughness correction, is applied to the threshold velocity. This factor is calculated as a function of grid-cell vegetation fraction based on Raupach (1992) as

$$r = \sqrt{1 - 0.5 x_f} \times \sqrt{1 + 100 x_f} \tag{18}$$

where $x_f$ is the frontal area index, calculated from the vegetation fraction ($c_f$) as

$$x_f = 0.35 \times \ln\left(1 - c_f\right) \tag{19}$$

Vegetation fraction ($c_f$) is stored in the model as the variable *greenfract*, and is determined from the MODIS Fraction of Photosynthetically Active Radiation (FPAR) absorbed by green vegetation monthly climatological values. This correction factor has a substantial impact on the threshold friction velocity. For example, a vegetation fraction of 0.2 (20% vegetation

coverage) results in a near tripling of the threshold friction velocity. We will see in our results below that this correction factor is a leading cause of differences in dust emission between the AFWA and UoC models.

Once the corrected threshold friction velocity ($u_{*t}\left(d, \theta_v, r\right)$) is determined, the calculation of saltation fluxes for each particle size bin, based on wind speed, is very similar in the UoC and AFWA schemes, though UoC uses more size bins (100 vs. 9 as AFWA is currently implemented). The UoC scheme uses a saltation flux equation that is very similar to the one used in

the AFWA scheme (Eq. (10)), with minor adjustments. This is presented here as Eq. (20). Note that until a bug fix released in January 2018, there was an error in the implementation of this equation, which is discussed below.

$$q\left(d\right) = \begin{cases} \left(1 - c_f\right) 2.3 \frac{\rho_a}{g} u_*^3 \left(1 - \frac{u_{*t}(d,\theta_v,r)}{u_*}\right)\left(1 + \frac{u_{*t}(d,\theta_v,r)}{u_*}\right)^2, & u_* \geq u_{*t}\left(d, \theta_v, r\right) \\ 0, & u_* < u_{*t}\left(d, \theta_v, r\right) \end{cases} \tag{20}$$

The two differences in comparison with the AFWA scheme are (1) an adjustment for vegetated fraction of the surface $\left(1 - c_f\right)$ and (2) the factor of 2.3, which replaces the empirical proportionality constant in the AFWA model ($C_{mb}$). In the

AFWA scheme, this constant is set to 1.0 as suggested by Marticorena et al. (1997) and Darmenova et al. (2009). The UoC value of 2.3 is closer to the value used in the original MB95 approach of 2.61 in accordance with findings by White (1979). The remainder of the equation, documented in S11, is identical to that used in the AFWA scheme. We note below, in section 3.3.2, however, that the implementation of this equation and the vegetation correction factor $\left(1 - c_f\right)$ in some versions of the code is not exactly as documented in the S11 paper, resulting in an important difference in model behaviors between AFWA

and UoC.

In all UoC schemes, just as in AFWA, the saltating particle load in each size bin is also dependent on the fraction of the parent soil consisting of particles in that size bin, and on the erodibility of soil at that location. Soil erodibility is again handled using the dust source strength parameterization (stored as variable *EROD*) from the original GOCART function (Eq. (4)). Here,





however, erodibility is treated as a binary. The binary source function is denoted ($S_b$) and set to 1 anywhere source strength is greater than 0. The parent soil particle size distribution is incorporated by multiplying the uncorrected (i.e. theoretical wind based, not supply limited) saltation flux for each bin $q(d)$ by a term representing the availability of saltation particles. The resulting saltation flux equation is

$$Q(d) = q(d)\, p(d)\, S_b \tag{21}$$

where the calculation of the particle availability term $p(d)$ treats free soil particles and particles contained in aggregates as separate categories. This is in contrast to the AFWA scheme, which handles all soil particles according to a single fundamental particle size distribution (see Eqs. (11) and (12)) and addresses the relative surficial area coverage of each particle class rather than handling them based on a bulk particle fraction. The term capturing the fraction of the soil consisting of available saltation particles in a given category is labeled as variable *dpsds* in the code, and is calculated according to Eq. (22) (equivalent to Eq. (8) in S11):

$$p(d) = \gamma \times p_m(d) + (1 - \gamma) \times p_f(d) \tag{22}$$

where $p_m(d)$ and $p_f(d)$ represent the minimally and fully disturbed particle size distribution (specifically, the array of the particle size fractions within diameter bin $d$), and where $\gamma$ is a function describing how easily released aggregated particles are. The values of $p_m(d)$ and $p_f(d)$ are set based on soil maps, as described below. Limitations in the quality of the input data potentially have large impacts on model results. In the S01 and S04 sub-options, the value for $\gamma$ is calculated based on an assumption that higher wind speeds can better break up aggregates (e.g., Alfaro et al., 1997) according to

$$\gamma = \exp\left[-k_1 \left(u_* - u_{*t}(d)\right)^3\right] \tag{23}$$

where $k_1$ is a constant equal to 1, $u_*$ is the friction velocity, and $u_{*t}(d)$ is the threshold friction velocity from Eq. (17), prior to correction for soil moisture and ground cover. Equation (23) here is replicated from Eq. (7) of S04 and Eq. (17) of S01. Field observations presented in S11 suggested the impact of wind speed on the released *dust* particle size is not significant, and so the S11 sub-option sets the value of $\gamma$ to 1, simplifying the *dust* emission parameterization. The S11 paper does not, however, address whether this simplification applies also in the calculation of size-resolved *saltation flux*. In the S11 code, $\gamma$ is calculated as in Eq. (23) for all UoC sub-options, such that $\gamma$ factor is the same as the S01 and S04 sub-options in calculation of saltation flux.

Once the saltation fluxes are calculated, the next major step in the model is calculating dust emission flux from the saltation flux, ($q(d_s)$). This step is comparable in function to the much simpler Eq. (14) in the AFWA scheme. The more sophisticated UoC models predict dust emission in each dust size category caused by saltating particles in each saltation size category (see Eq. (52) in S01 and Eq. (6) in S04), as opposed to calculating a single bulk dust emission mass from the effects of




all saltating particle classes and then apportioning this bulk emission into dust size bins with a fixed parameterization. The complex parameterization takes into account the particle size distributions of both the parent soil dust and saltation particles. This calculation is where the S01, S04, and S11 model sub-options differ most, and we will briefly present each sub-option parameterization.

S01 derives and uses the most complex form of the parameterization, described as Eq. (52) in S01. The parameterization includes effects of soil particle aggregation, parent soil particle size distribution, saltating particle size distribution, and soil plastic pressure, among other tuning parameters.

$$F\left(d_i, d_s\right) = c_y \left[(1 - \gamma) + \gamma \sigma_p\right] \frac{q\left(d_s\right) g}{m u_*^2} \left(\rho_b \eta_{f,i} \Omega + m \eta_{c,i}\right) \tag{24}$$

where $c_y = 0.00001$ is a dimensionless constant, $\gamma$ is evaluated as in Eq. (21), $\eta_{f,i}$ and $\eta_{m,i}$ are, respectively, the fully- and
minimally-disturbed dust fraction in bin $d_i$, $\rho_b = 1000$ kg m$^{-3}$ is the assumed bulk density of the soil, $\eta_{c,i}$ is the fraction of soil available for disaggregation ($\eta_{f,i} - \eta_{m,i}$), $\sigma_p = \frac{\eta_{f,i}}{\eta_{m,i}} = \frac{p_f(d_i)}{p_m(d_i)}$, $m$ = mass of the particle, and $g$ is the gravitational constant in m s$^{-2}$. The term $\Omega$ represents the efficiency of dust emission from bombardments or collisions and is implemented in the model after Lu and Shao (1999) as

$$\Omega = \frac{m U_p^2}{2 \varrho d \beta_v^2} \left(\sin\left(2\alpha_i\right) - 4\sin^2\left(\alpha_i\right)\right) + \frac{7.5\pi}{d} \left(\frac{U_p \sin\left(\alpha_i\right)}{\beta_v}\right)^3 \tag{25}$$

where $U_p$ is the impact velocity, $\beta_v = \sqrt{\frac{2\varrho d}{m}}$, $\varrho$ is soil plastic pressure, $\alpha_i$ is the incidence angle of the collisions, $m$ is the particle mass, and $d$ is the particle diameter.

S04 simplifies the scheme for estimating the dust emission from saltation collisions by fixing several of the free variables in Eq. (25) which were not readily available in measurements, including setting the collision angle to 15 degrees, setting $U = 10u_*$, and setting the particle density to 2.6 times the soil bulk density. This allows a revised form of the equation for
bombardment efficiency to be derived which is particle size independent

$$\sigma_m = 12 u_*^2 \frac{\rho_b}{\varrho} \left(1 + 14 u_* \sqrt{\frac{\rho_b}{\varrho}}\right) \tag{26}$$

where $u_*$ is the friction velocity, $\rho_b = 1000$ kg m$^{-3}$ is bulk soil density, and $\varrho = 30000$ N m$^{-2}$ is the soil plastic pressure. We note, in particular, the very strong role that soil plastic pressure plays in the emission through this term, and further note that the value for soil plastic pressure is set to a constant in the model implementation, despite being a parameter well known to be
subject to variations with soil type. Incorporating $\sigma_m$ into the dust emission flux equation and simplifying results in Eq. (27); the revised flux equation used by S04 (S04 Eq. (6))

$$F\left(d_i, d_s\right) = c_y \eta_{f,i} \left[(1 - \gamma) + \gamma \sigma_p\right] \frac{q\left(d_s\right) g}{u_*^2} \left(1 + \sigma_m\right) \tag{27}$$





S11 further simplifies the scheme by calculating dust emission based on a single integrated saltation flux, rather than based on fluxes of saltating particles in each individual saltation bin (and setting $\gamma = 1$ as noted above in the discussion of Eq. (21)). Dust emission is then calculated for each dust size bin according to Eq. (28) (S11 Eq. (34))

$$F(d_i) = c_y \eta_{m,i} \frac{gQ_{total}}{u_*^2}(1 + \sigma_m)$$ (28)

where $c_y = 0.00001$ is a dimensionless coefficient, $\eta_{m,i}$ is the fraction of dust in size bin $i$ that is free in minimally disturbed soil, $\sigma_m$ is the bombardment efficiency, $g$ is the gravitational constant, $Q_{total}$ is the saltation flux, and $u_*$ is the friction velocity. Total saltation flux $Q_{total}$ is calculated by integrating across all particle size bins using Eq. (29) (S11 Eq. (20))

$$Q_{total} = \sum_{d=1}^{\#bins} Q(d)$$ (29)

This S11 approach is similar to the AFWA scheme, which integrates saltation flux across all saltation particle size bins (Eq.
(13)) and calculates a total dust emission from a total integrated saltation flux (Eq. (14)). The two models differ in that the AFWA scheme sums the mass of all dust fluxes and then apportions the dust into size fractions based on a breaking function (Eq. (15)). The simplified S11 scheme however, allows the dust particle size distribution to be based on parent soil type (Eq. (28)).

In S01 and S04, the size-resolved dust emission is calculated by integrating dust emissions of each dust bin over all saltation
bins. During this step, an additional factor of $1 - c_f$ is applied. This factor does not appear in the papers that document these schemes (S01, S04, S11) and may be in error.

$$F(j) = (1 - c_f) \sum_{i=1}^{bins=100} F(i,j)$$ (30)

The S11 scheme yields size-resolved dust emission $F(j)$ directly, but the factor of $1 - c_f$ is also applied before emissions are reported to the atmosphere model

$$F(j) = (1 - c_f) F(j)$$ (31)

In all UoC schemes, the total dust emission, $F_{total}$, is calculated by integrating over all emissions bins.

$$F_{total} = \sum_{j=1}^{bins=dust} F(j)$$ (32)





### 3.3.1 Impact of soil data on the UoC scheme

The effect of the more sophisticated approach in the UoC schemes is to make both the saltating and emitted dust particle size distributions sensitive to parent soil particle size distribution in S01 and S04 and to make the emitted dust particle size distribution sensitive to parent soil particle size distribution in S11. The approach makes the UoC parameterization schemes the most

physically-based of the WRF-Chem dust emission schemes. Input data limitations restrict the benefit of these sophisticated parameterizations, however. Measurements of these soil characteristics are generally unavailable, particularly over mesoscale domains (on the order of 10km grid spacing), an issue noted in the Shao publications. For example, the degree of soil aggregation, used in the UoC schemes as the fully-disturbed and minimally-disturbed soil particle size distribution, is not widely measured or widely available in soil databases, nor is the soil plastic presuure. Within WRF-Chem the soil plastic pressure is

simply set to a constant and must be tuned to match local soil conditions. The particle size distributions are derived based on a conversion between the soil particle size information for the surface layer of soil (0–30cm) originally derived from the Food and Agriculture Organization (FAO) digital Soil Map of the World (SMW) by Reynolds et al. (2000) and a series of 12 soil modes described in S04 (this is carried out in subroutine $h\_c$). The original 5-minute grid of the FAO-SMW map is interpolated to a 1km grid for use in this application. The soil type indicated in the FAO-SMW map is converted to its fully-disturbed and

minimally-disturbed particle size distributions by compositing the several modes, each containing log normal particle distributions with differing coefficients (e.g., see S01 Eq. (54), S04 Eq. (15), and S11 Eq. (21)). We note that the number and character of the soil modes being composited varies in the Shao publications from 3 (S01 Table 2) to 12 (S04 Table 1) to 4 (S11 Table 2). All three model sub-options, however, are implemented using the 12-mode soil mixing.

Dependence on the other key soil parameter, soil plastic pressure, controls the mass ejected during bombardment collisions.

In the Shao papers, test cases are run to determine the best fit for the soil plastic strength based on observational dust emission data, along with a dimensionless tuning coefficient $c_y$. Data presented in S04 indicates that soil plastic pressure varies over roughly 2 orders of magnitude from 500 to 50000 Pa for sandy to clay rich soils, respectively (see S04, Table 3). Similarly, the tuning constant $c_y$ is found to vary from 1e-5 to 5e-5 (it is set to 1e-5 by default in the model). A serious model limitation in terms of running the UoC scheme at mesoscale is that the value of the soil plastic pressure is set to a single value domain-wide

and does not vary with soil type. Given that the value varies so widely over various soil types, mismatch in part of the domain is likely. The default for this value in WRF-Chem versions 3.6.1–4.0 is set to 30,000 Pa, appropriate for clay-rich soils according to S04.

### 3.3.2 Differences between UoC literature documentation and code

Similar to the GOCART-WRF scheme, we note that there are several discrepancies between the code realization in WRF-Chem

and the documentation published in the literature in S01, S04, and S11. Again, we document these here for the benefit of the community:

1. The equation used to calculate the saltation flux $Q$ implemented in the WRF-Chem code versions 3.2–3.9.1 was subtly, but significantly, different from the equation documented in S11, Eq. (19). This error was corrected in a bug fix in




early 2018, and this correction is now included in all WRF-Chem versions. Specifically, the equation provided in S11 computes the saltation flux for each saltation particle size bin, $q_i$, as:

$$q_i = \begin{cases} (1 - c_f) \, 2.3 \frac{\rho_a}{g} u_*^3 \left(1 - \frac{u_{*t}(i)}{u_*}\right)\left(1 + \frac{u_{*t}(i)}{u_*}\right)^2, & u_* \geq u_{*t}(i) \\ 0, & u_* < u_{*t}(i) \end{cases}$$

(33)

where $c_f$ is the vegetation fraction, $g$ is gravity, $u_*$ is the friction velocity, and $\rho_a$ is air density. We noted above that the form of this equation is nearly identical to the equation used in the AFWA scheme (Eq. 10), with both ultimately derived from work by Kawamura (1951). Notably, the implementation in all UoC code versions implemented in WRF-Chem prior to the bug fix released 9 January, 2018 treats the final term as:

$$\left(1 + \left[\frac{u_{*t}(i)}{u_*}\right]^2\right)$$

(34)

Changing the order of operations from how it is documented in S11:

$$\left(1 + \frac{u_{*t}(i)}{u_*}\right)^2$$

(35)

Given reasonable friction velocities, the effect could change the saltation flux by a factor of two or more, resulting in substantial impacts on output.

2. The equation used to calculate the threshold friction velocity for particles in each saltation bin size, $u_{*t}(d_s)$, is referenced as originating from Eq. (21) in Shao and Lu (2000) by S01 and S11. The equation given in Shao and Lu (2000) is

$$u_{*t}(d) = \sqrt{A_N \left(\sigma_p g d + \frac{\gamma_c}{\rho_p d}\right)}$$

(17, repeated)

where $A_N = 0.0123$, $\sigma_p$ = the ratio of particle to air density, $g$ is the gravitational constant, $d$ is particle diameter, and $\rho_p$ is the particle density. The coefficient $\gamma_c$ is set to 1.65 x $10^{-4}$ kg s$^{-2}$ in the code, while it is specified as 3.0 x $10^{-4}$ kg s$^{-2}$ in Shao and Lu (2000).

3. The implementation of the code appears to include the vegetation coverage correction factor, $1 - c_f$, used in the saltation flux calculation above twice (in addition to the use of this term in calculating the surface roughness correction factor). The first time it is included is directly in the calculation of the saltation flux, which is carried out using Eq. (20). The factor is again applied during the integration of the dust emissions across the dust and saltation size bins (Eqs. (30) and (31)).





4. The documentation for the earlier UoC models (S01 and S04) indicates they use different equations for calculating saltation flux based on current wind speed and threshold velocity than that used in S11. These equations are of similar form and would produce similar saltation flux output to what would be produced by the equation described in S11 (see S01 Eq. (23), which is derived from Owen (1964)). We find no evidence, however, that these separate means of calculating saltation flux are actually implemented in the S01 and S04 sub-options of the model code. It appears that all three sub-options are currently using the saltation flux presented in Eq. (20) and described above.

5. We note that the number and character of the soil modes being composited to determine the free dust fraction at particle sizes varies in the Shao publications from 3 (S01 Table 2) to 12 (S04 Table 1) to 4 (S11 Table 2). As implemented in the model, the 12-mode soil mixing of S04 applies to all three UoC sub-options.

6. The formula for the emission of dust during particle collisions implemented in the S01 sub-option differs from the from that documented in Eq. (36) of S01, however, following this equation back to its source as Eq. (8) in Lu and Shao (1999) shows that the implementation matches the original source, and the error is in documentation in S01. To illustrate the difference clearly, the implemented equation, described as Eq. (25) above (and reproduced here), is presented alongside the version documented in Eq. (36) of S01 (also presented here as Eq. (36) by coincidence). There are several key differences in the order of operations and the initial factors differ by $1/d$. For example, the cubed power is applied only to the rightmost $sin$ function, rather than to the group of terms associated with it, and the leftmost factor is multiplied by all other terms in Eq. (36), but only by the center two $sin$ terms in Eq. (25).

$$\Omega = \frac{mU_p{}^2}{2\varrho d\beta_v{}^2}\left(\sin\left(2\alpha_i\right) - 4\sin^2\left(\alpha_i\right)\right) + \frac{7.5\pi}{d}\left(\frac{U_p\sin\left(\alpha_i\right)}{\beta_v}\right)^3 \qquad \text{(25, repeated)}$$

$$\Omega = \frac{\pi\rho_p d^3 U_p{}^2}{2\varrho}\left(\sin\left(2\alpha_i\right) - 4\sin^2\left(\alpha_i\right) + \frac{7.5\pi U_p sin^3\left(\alpha_i\right)}{\beta_v d}\right) \qquad \text{(36)}$$

## 3.4 Synopsis of key differences between UoC and AFWA schemes

1. The original derivation of the UoC model handled aerodynamic entrainment, saltation bombardment, and aggregate disintegration mechanisms separately (see derivation of Eq. (52) in S01, Section 5), as opposed to handling all dust emission in a single bombardment-like process as is done in the AFWA scheme.

2. The UoC model scheme for calculating dust emission flux from saltation flux (e.g., captured by Eq. (52) in S01, Eqs. (6), (7), and (11) in S04, and Eqs. (11) and (34) in S11) depends on relatively sophisticated knowledge of the parent soil, including the soil particle size distribution (the only term which the AFWA scheme also depends on), measures of the degree of soil disturbance (e.g., captured in $\sigma_p$, Eq. (7), S04), and the soil bonding, presented as the soil plastic pressure, which controls the mass ejection caused by saltation bombardment (e.g., captured in $\Omega$ in S01 and in $\sigma_m$ in S04 and S11).





The degree of this dependence on sophisticated soil properties decreases in the more simplified S04 and S11 schemes. For example, part of the dependence of aggregate breakdown on wind speed is removed in the S11 simplification based on field observations that indicated no wind speed dependence. The dependence of emission on soil plastic pressure and on the free soil particle size distribution, however, is common to all three, and the values of these parameters have substantial influence over model output.

3. The UoC model incorporates a correction factor in the calculation of saltation flux for soil vegetation coverage. This factor has modest impacts on results, and our test case indicates its utility may suffer from low quality input data.

4. The UoC model incorporates a second correction factor in the calculation of threshold friction velocity for soil surface roughness, which is determined from the soil vegetation coverage layer.

## 4 Test case model configuration

### 4.1 Model and domain setup

We use the Weather Research and Forecast with Chemistry model (WRF-Chem) version 3.8.1 to simulate the emission and transport of dust in our test cases with each of the three default dust emission schemes (Grell et al., 2005; Fast et al., 2006; Skamarock et al., 2008). The model domain for this test is bounded by corner points at approximately SW = [7.9 °N, 16.5 °E]; NW = [51.8 °N, 11.6 °W]; SE = [10.0 °N, 62.4 °W]; NE = [56.8 °N, 85.2 °E], is configured with 484x417 grid points on a horizontal grid spacing of 12km, and is shown in Fig. 2. The vertical grid contained 48 levels and followed a stretched sigma-coordinate that favored higher vertical resolution near the surface. Initial and lateral boundary conditions were forced using the Global Forecast System Final Analysis (GFS-FNL) 6-hourly, 1-degree resolution reanalysis product NOAA/NCEP (2000). The simulation was performed over the five-day period between 22 January 2010 and 27 January 2010, but the first 36 hours of the simulation were disregarded as spin up to allow the model to adjust to the initial and lateral boundary conditions.

Atmospheric dust was initialized using a "cold start" approach (i.e., the dust concentration in the atmosphere is initialized as zero everywhere). The model background chemistry for other aerosol species was generated using the GOCART simple option in WRF-Chem. Background sea salt emissions were based on the lowest model level wind speeds over the oceans (Gong, 2003), and the other background non-dust aerosol emissions within the domain were set using the PREP-CHEM-SRC preprocessing software (Freitas et al., 2011) using the GOCART climatological emission datasets. No aerosols were transported into the domain across the lateral boundaries during the simulations – a reasonable approximation given that we were primarily concerned with large localized dust emission events far from the domain boundaries. Importantly, the aerosol radiative feedbacks were turned off. Therefore, modeled aerosol concentrations had no impact on the model meteorology, ensuring a simple comparison of dust emission schemes under identical forcing. A full description of the model configuration, including chemistry and physics parameterizations, is presented in Table 5.

The dust emission parameterizations are the main focus on this paper and are discussed separately above in Section 3. Each of the three standard dust emission parameterizations covered is tested, and we compare the results below. All three dust





emission schemes tested were run in the "default" configuration supplied with WRF-Chem version 3.8.1 release to permit the most straightforward comparison, with all constants set as supplied in the code release and described above in documentation for each model. For the purposes of this paper, we chose to make comparisons to the moderately simplified model version of the UoC scheme described in S04.

5    For inter-comparison of model results with remote sensing data, model atmospheric extinction coefficients are calculated for the 550nm wavelength using the WRF-Chem optics routines (Barnard et al., 2010). Model simulated AOD is then calculated by vertically summing the extinction coefficient throughout the atmospheric column

$$AOD = \sum_{k=1}^{n_k} \mu_{550,k} \Delta z \tag{37}$$

where $k$ is the model vertical level, $\mu_{550}$ is the extinction coefficient at 550nm, and $\Delta z$ is the physical depth of each vertical 10   level.

Integrated column AOD is sampled from the model for comparison with satellite remote sensing observations collected from the Cloud-Aerosol Lidar with Orthogonal Polarization instrument (CALIOP) at the grid point nearest to observational geographic coordinates (Lat/Lon). For comparison with CALIOP data, coordinates used represent the midpoint of the 15 along-track samples that are averaged to produce a single AOD estimate. Since samples are collected every 333m by CALIOP, actual 15   observations extend 2.5km from the midpoint in each direction along track.

### 4.2  Description of selected test event

The test event selected for our emission scheme inter-comparison was a dust storm in southwest Asia forced by a large scale synoptic event. We chose this location because we expect that the conditions the AFWA scheme was created for frequently prevail there. Specifically, spurious dust lofting under light wind conditions has been noted in this region in WRF-Chem 20   runs with the GOCART-WRF dust emission scheme activated, as discussed in section 3.2. The atmospheric dust observed by satellite remote sensing platforms during this event originated largely in Western Iraq and Syria.

While we compare remote sensing and model results throughout the event, we focus most of our analysis on the time period between 0600 UTC and 2300 UTC on January 25th when a classic wintertime Shamal moved across the analysis domain, causing emission and lofting of dust from the Syrian Desert. During a Shamal, a cold front sweeps across the Arabian Peninsula 25   allowing a high pressure to build in from the northwest and strengthen across Saudi Arabia. The synoptic pattern forces strong northwesterly surface winds to blow across the Syrian Desert and, often, lofts large quantities of dust into the atmosphere.

We characterize the synoptic evolution and evaluate the meteorology of the WRF-Chem simulation using the Climate Forecast System Reanalysis product (CFSR; Saha et al., 2010). The CFSR product combines the Climate Forecast System coupled ocean/atmosphere model reforecast data with an assimilation of available observations, including data from surface, 30   radiosonde, aircraft, and satellite observations. Critically, this reanalysis dataset is independent of the GFS-FNL reanalysis dataset used to force the WRF-Chem model, increasing the independence of this model evaluation. We specifically utilize 700hPa geopotential height, 850hPa temperature, and 925hPa winds for the comparison. These variables provide a good visu-




alization of the synoptic forcing, identify frontal boundaries, and illustrate large-scale low-level wind patterns. Figure 3 shows snapshot images of these variables over the analysis domain. Prior to the event, at 0000 UTC on 24 January 2010, low-level southerly winds were present across much of the Arabian Peninsula, advecting warm air from the south, and a mid-level trough of low pressure was present to the northwest of the region (Fig. 3a). By 0000 UTC on 25 January 2010, the mid-level trough
dropped south onto the Syria / Turkey border, and a cold front moved into Iraq initiating the dust event (Fig. 3b). By 1200 UTC on 25 January 2010, the front entered Iran, and strong westerly winds covered much of the Syrian Desert (Fig. 3c). It was at this time that a large dust plume was visible across the Syrian Desert centered along the Iraq / Saudi Arabia border in remotely-sensed imagery (Fig. 4). At 0000 UTC on 26 January 2010, the front was weakening as it pushed south across Saudi Arabia, and a secondary cold front was moving south into northern Iraq and Syria (Fig. 3d).

To evaluate the realism of the modeled synoptic evolution, we compared the variables used to characterize the synoptic environment from WRF-Chem (Fig. 3a–d) with the independent CFSR data (Fig. 3e–h). The synoptic evolution produced by the WRF-Chem model was very similar to the one in the CFSR, indicating that WRF-Chem performed adequately in simulating the meteorology. Further comparisons to radiosonde data (not shown) indicated WRF-Chem was able to adequately reproduce the observed atmospheric wind and temperature profiles (Letcher and LeGrand, in press). Importantly, WRF-Chem was able to
reproduce the observed boundary layer winds quite well over the dust source region, a critical requirement to accurately simulate dust emission. The general consistency of the modeled and observed meteorology indicates that discrepancies between modeled and observed dust in the atmosphere are largely attributable to the simulated dust emissions, rather than to the simulated meteorology. Additionally, each of the three model simulations experience the same meteorology, such that differences between the modeled dust emissions can be entirely attributed to the emission parameterizations.

## 5   Validation data access and processing

### 5.1   MODIS imagery (truecolor and dust-enhanced products)

We utilize 1km-resolution truecolor and dust-enhanced satellite-imagery derived from MODIS data to qualitatively assess the general origin and extent of the dust plumes. Image dust-enhancement was performed using a processing algorithm by Miller (2003), in which atmospheric dust is distinguished from the underlying background terrain using visible, near infrared, thermal
infrared, and water vapor channels. Lofted dust appears pink, landscapes have blue and green hues, water and steep terrain are red, and clouds appear aqua or cyan in the resulting image. The script used for acquiring MODIS granules and generating imagery in GeoTiff format is available in Sinclair and Jones (2017).

### 5.2   CALIOP data

We use version 4 (V4) of the level 2 (L2) vertical feature mask data product (CAL_LID_L2_VFM-Standard-V4-1) from
the Cloud-Aerosol Lidar with Orthogonal Polarization instrument (CALIOP) on board the Cloud-Aerosol Lidar and Infrared Pathfinder Satellite Observation (CALIPSO) mission to identify atmospheric aerosol observed in the modeled domain (Winker,



2009). These data provide an along-track record of cloud and aerosol layers observed by the CALIOP lidar averaged over 5km bins (15 profiles at 333m spacing), which classifies observations as clean air, clouds, aerosols, stratospheric features, surface, subsurface, and totally attenuated backscatter (no signal). In addition, nine aerosol subtypes (clean marine, dust, polluted continental/smoke, clean continental, polluted dust, elevated smoke, dusty marine, volcanic ash, and others) are derived in the

V4 L2 Aerosol Layer product (CAL_LID_L2_05kmAPro-Standard-V4-10). These are used to verify that aerosol clouds being investigated in this study are primarily dust. We obtain observations of aerosol extinction profiles from the V4 L2 Aerosol Profile product (CAL_LID_L2_05kmAPro-Standard-V4-10, Young and Vaughan, 2009), which are compared directly to the modeled atmospheric extinction profiles. CALIOP AOD is obtained by integrating over the vertical extinction column. All products are available through the NASA data portal at search.earthdata.nasa.gov.

## 6  Results

Results from the three dust schemes (Fig. 5–7) demonstrate substantial differences in outcomes between the GOCART-WRF scheme (*dust_opt=1*), and the other two schemes (AFWA and UoC). Smaller but still substantial differences exist between the AFWA and UoC schemes (*dust_opt=3* and *dust_opt=4*). Figure 4 shows MODIS truecolor and dust-enhanced imagery of the peak dust emissions. The extent of the dust cloud can be seen to imply emissions encompassing the Syrian desert

region in Jordan, Syria, and Western Iraq. Figure 5 shows modeled aerosol optical depth at 550nm for each of the three dust schemes, at six snapshots in time during the event coinciding with CALIOP overpasses. CALIOP-derived AOD transects (left-most line) are overlain on the plots adjacent to equivalent model-derived AOD transects for comparison (right-most line). A representation of CALIOP observed clouds is also shown to indicate pixels with suspect AOD observations (center line). Figure 6 shows full vertical curtains of aerosol extinction profiles along the CALIOP transects for each of the 6 overpasses

from CALIOP observations (row 1) and the model outputs (row 2–4). Finally, Fig. 7 shows the dust emissions derived for each of the emission schemes, at time snapshots representing three CALIOP overpass times and three other times during the dust emission event.

The collection of these model outputs clearly demonstrates that the GOCART-WRF scheme produces the largest atmospheric dust content, and that the dust lofts from across the widest area, including intense emissions from the Syrian desert in eastern

Syria, Jordan, and Western Iraq and lower intensity emissions in the Northern Arabian desert areas of southern and western Iraq and northern Saudi Arabia (Fig. 7). The dust emissions occur over a wider area and continue temporally longer than they do in the other schemes, including in areas experiencing lower wind speeds. This outcome is consistent with the spurious dust lofting noted by earlier works. The result of these large-scale emissions is substantial AOD over large areas of the model domain (Fig. 5). The excessive area experiencing dust lofting is largely expected given the treatment of the threshold wind speed discussed

in section 3.2.

The AFWA and UoC schemes both produce much more localized emissions and emit dust only under the higher wind conditions present early on January 25 (Fig. 7). Emissions in the AFWA scheme originate from the Syrian desert in southern and eastern Syria, western Iraq, and eastern Jordan, but are limited beyond this domain, and of much lower intensity than seen





in the GOCART-WRF scheme. These result in AOD patterns that mirror the "pulse" of dust emission as the front passes over the Syrian desert. The pulse is then advected eastward and northward out of the model domain (Fig. 5). The spatial configuration of emissions is much more localized for the UoC scheme, restricted to intense emission sites in the Syrian desert, primarily in southern Syria, but also in extreme eastern Jordan and extreme western Iraq. The modeled AOD resulting from the localized

emission of the UoC scheme is then an intense pulse with relatively hard boundaries. Similar to the AFWA scheme, this is advected east and northward out of the domain, but covers a much smaller spatial extent during this time.

Compared to the spatial extent of the AOD plume seen in the MODIS observations (Fig. 4), the modeled AOD in the AFWA scheme (Fig. 5) produces the best match to the AOD seen in the cloud free region within the MODIS observations, in this particular test case. Modeled AOD shows too small a spatial extent in the UoC scheme and too large a spatial extent in the

GOCART-WRF scheme (Fig. 5). This single test case comparison does not imply that any of the three models is superior in all cases. This result, however, provides the basis for investigating the reasons for the particular model behaviors in the discussion that follows.

More detailed comparisons of modeled and observed dust in the atmosphere are presented using the CALIOP lidar data. Total column AOD is presented along the CALIOP tracks in Fig. 5. The parallel transects represent the observed (left) and

modeled AOD (right) with cloud cover that restricts CALIOP retrieval of full column AOD indicated in the center transect. Note that observed and modeled AOD should only be compared in areas not impacted by cloud cover. Unfortunately, high observed AOD frequently occurs in close proximity to cloud cover and none of the available CALIOP transects directly sample the main dust plume of this event near the time of peak emissions. While these limitations hinder a robust comparison, a general result is that the GOCART-WRF scheme tends to produce higher AOD along the CALIOP transect than observations

show (e.g., Fig. 5, row 7), while the AFWA and UoC schemes both show more limited AOD along the transects which appear smaller in extent than suggested by observations. All models appear to under predict the highest values of observed AOD. Closer examination of this is needed in profile format to better assess agreement.

Modeled and observed aerosol extinction profiles are presented in Fig. 6. A combined plot representing several CALIOP observations is presented in the first row. The plot is based on vertical feature mask data (to show clouds) and extinction

profiles, where available. Optically thick clouds are masked in light gray and area underneath optically thick clouds (no data) is masked in dark gray. This more clearly shows the substantial limitations on available data in the lower atmosphere imposed by cloud cover and the reason for limited observations of high total column AOD in the transects shown in Fig. 5. The extinction coefficients presented, both in this observed data and in the model profiles below may be reasonably thought of as being caused entirely by dust, because aerosol extinction is overwhelmingly attributed to mineral dust in both CALIOP Aerosol

Layer Product and in modeled data.

The modeled extinction profiles presented in rows 2–4 indicate that the location of dust in the atmosphere is largely consistent between models, but that the amount of dust in the atmosphere differs substantially between models, with the most dust in the GOCART-WRF scheme and the least dust in the UoC scheme. The altitude and spatial placement of the modeled atmospheric dust (as indicated by extinction coefficients) along CALIOP passes collected 24 Jan 2010 11:00 UTC, 24 Jan 2010 23:00 UTC,

26 Jan 2010 00:00 UTC, and 26 Jan 2010 01:00 UTC all appear consistent with observations, though the observed atmospheric





extinction is higher than the amount present in all models. In these, the overall dust entrained into the atmosphere in the GOCART-WRF scheme, even though it is emitted from far too large a spatial area, is the best match for observed extinction profiles, in terms of magnitude. Limited observations due to cloud cover make the 25 Jan 2010 10:00 UTC pass challenging to assess. Modeled dust on 26 Jan 2010 23:00 UTC is consistent with the other four time steps, in that altitude and spatial

placement of the model dust (extinction coefficients) along the southern end of the transect broadly matches observations, but differs in that the GOCART-WRF and AFWA schemes exhibit much stronger extinction profiles in the central part of the transect from 32.5 °N to 27.5 °N, than are shown in observations. We summarize these results by noting that the overall amount of entrained dust appears to be too low in all models, and that the spatial extent of the emissions are too large in the GOCART-WRF scheme, too small in the UoC scheme, and broadly similar to observations in the AFWA scheme.

**7   Discussion**

We primarily intend our test-case data to be a tool to discuss the differences between the three WRF-Chem dust emission schemes. We therefore explored the reasons for the differences between these emissions schemes in greater detail. Plotting several intermediate model variables as diagnostics illuminates the various sources of the large differences in the spatial extent and intensity of the modeled dust emissions and identifies highly sensitive parameters in the model. The intermediate model

variables are shown as a series of panels in Fig. 8 and 9, organized in the order the terms are used in the model calculations described above.

Here we were particularly interested in explaining the reasons for the differences in spatial coverage of dust emission in the UoC model scheme, relative to the AFWA scheme. Reasons for spurious dust lofting at low wind speeds in the GOCART-WRF scheme are well documented in our discussion in section 3.2, and by earlier papers (e.g., Colarco et al., 2003a) and require

little further investigation. In considering the UoC–AFWA differences, we first note that winds are high across the region where dust lofts in the AFWA model (Fig. 8, Row 1) – and largely equivalent in western Iraq and southern Syria, even though dust is only emitted in the Syrian portion of this area in the UoC model. The equivalent wind forcing across areas that do, and do not, emit dust within UoC suggests the difference is a fundamental part of the dust emission scheme. We hypothesized that this could be due to: (1) the source function ($S$ in the literature, or *EROD* in the model) being treated as a binary in UoC vs. as a

0–1 weighting factor in AFWA, (2) differences in calculated threshold friction velocity, especially related to the soil moisture correction and the roughness correction factor (which is applied only in UoC), and (3) the dependence of both saltation flux and dust emission calculations on the factor $1 - c_f$ in UoC, a factor which is not present in AFWA. We ultimately found that the restricted area of emissions is primarily due to the roughness correction factor (the second part of hypothesis 2), but that vegetation correction (hypothesis 3) and a coding error also play a role in the differences.

We tested these hypotheses by following the dust emission calculations through each of the three model parameterizations, showing intermediate factors in these calculations visually in Fig. 8 and 9, which represent the model state on 25 Jan 2010 at 1100 UTC, during the peak of dust emissions. We include the GOCART-WRF scheme for completeness, though we acknowl-



edge the attempt to make a step-by-step comparison with that model is imperfect because the GOCART-WRF scheme operates based on a direct relationship between wind speed and dust emission and does not track saltation sized particles separately.

We begin our analysis by calculating the dry soil threshold parameter to initiate particle mobilization for all three dust emission schemes (threshold velocity in the case of GOCART-WRF and threshold friction velocity for the AFWA and UoC schemes). In all models, the calculated dry soil threshold parameter is uniform (i.e., represented by a single value) everywhere the dust source function is nonzero, but has a value that differs between emissions schemes. Its value is represented by the uniform color shading on the maps in Fig. 8, row 2. Subsequent maps are built upon this using additional calculations. The AFWA and UoC schemes determine the dry soil threshold saltation friction velocity based on Eq. (5) and (17), respectively. Though the parameterizations are different, we note that the thresholds, shown for 60 μm particle size, are very similar between the UoC and AFWA schemes. We therefore conclude that minor differences in these threshold friction velocities are not a major cause of differences in dust emissions. For the GOCART-WRF scheme, we determine the dry soil threshold velocity is equal to 0.479 m s$^{-1}$ for a grain diameter of 16 μm (the effective diameter of the largest dust bin) using Eq. (5).

All three dust emission schemes include a correction for the threshold particle mobilization parameter based on the soil moisture. This correction factor is shown in Fig. 8, Row 3. The parameterization for calculating this correction in AFWA and UoC schemes is identical (Fécan et al., 1999), but we see slightly different output, presumably due to minor differences in coefficients applied to permit handling of moisture content in different units. As expected, these minor differences do not drive a significant difference in emitted dust mass. However, in comparing AFWA and UoC, a somewhat higher soil moisture correction is present across north central Saudi Arabia in the UoC scheme. This might cause a difference in dust lofting from that region under certain circumstances. In this case, neither model emits dust from this region (Fig. 7). The similarity in moisture correction factors leads to similar moisture-corrected threshold friction velocities for the UoC and AFWA schemes (Fig. 8, Row 4) leading us to reject the first part of hypothesis 2 and conclude that differences in moisture correction are not the principle cause of differences in emissions between AFWA and UoC in this case study.

The soil moisture correction parameterization in the GOCART-WRF scheme is quite different, and its value varies from 0 to 1.2, with values near zero for soils of very low moisture content. The values <1 effectively adjust the threshold velocity determined from the MB95 relationship downward, and thus this scheme treats the MB95-based threshold velocity as if it were valid for soil of moisture content 0.1, rather than as if it were for dry soil. In contrast, the adjustment in the AFWA scheme assumes MB95 velocities represent dry soil and adjusts the threshold friction velocity upward for higher moisture content. The behavior of the GOCART-WRF scheme, further reducing threshold velocities under dry soil conditions, is challenging to defend and likely further contributes to spurious low-wind dust lofting seen in the GOCART-WRF scheme (though the substitution of an equation intended for threshold friction velocities for 10m wind speeds discussed in Section 3.1 is a more important factor).

In the UoC scheme, the moisture-corrected threshold friction velocity is further modified by a roughness correction, calculated based on vegetation coverage (Eq. (19)). This factor is shown in Fig. 9, Row 1 in the UoC column. Ranging in value from 1 to 4, the factor substantially raises the threshold friction velocity over large parts of the domain. We note in particular that it is a strong candidate for being the primary cause of emissions reductions in Western Iraq, relative to those predicted by



the AFWA scheme, because it increases threshold friction velocity in Western Iraq by a factor of 2 or more, while southern Syria remains near 1. There is no step in the AFWA or GOCART-WRF schemes that is broadly comparable to the roughness correction in UoC. We note that there is an optional run time flag in the AFWA model that would allow a user to feed in a vegetation mask through an auxiliary channel, but this is not used as part of the default configuration.

Threshold friction velocities with all corrections applied are then shown in Fig. 9, Row 2. These friction velocities, which can be compared against those which have only the moisture correction applied (in Fig. 8, Row 4) clearly show that the vegetation correction increases the threshold friction velocity across the western Iraq area in the UoC scheme, while leaving the threshold friction velocity similar to the AFWA scheme in southern Syria.

Theoretical saltation flux is next calculated from the WRF-Chem simulated wind speed and friction velocity and the threshold
friction velocity. This is shown in Fig. 9, row 3 for particles of 60 μm size (AFWA and UoC) and 16 μm size (GOCART-WRF). UoC and AFWA use the same equation to derive saltation flux, with minor modifications of factors (Eqs. (9) and (16)) and a code implementation error in the UoC scheme (see Section 3.2.3 for discussion). The minor modification of factors, namely the addition of a $(1 - c_f)$ factor and the adjustment of a constant factor from 1 to 2.3 in UoC relative to AFWA, should generally result in increased saltation flux in UoC for locations having equivalent corrected threshold friction velocities in Fig. 8 Row 2,
but by no more than a factor of 2.3. The UoC code implementation error in Eq. (16), however, more than counteracts this, and results in substantially lower theoretical saltation flux than would be expected (by about one order of magnitude). The result is that saltation fluxes within the (limited) areas having similar threshold friction velocities is lower in UoC, relative to the AFWA model. Releases with the bug fix announced in early 2018 should be expected to produce slightly higher emissions from UoC relative to AFWA under conditions where both models produce similar threshold friction velocities. This would help improve
the overall emission of dust in the UoC scheme (which was too low) but would not impact the limited spatial extent of dust emissions which we seek to understand.

Theoretical saltation fluxes are converted to predicted saltation fluxes by considering the availability of erodible substrate, which is captured in all schemes by the source strength function, though the manifestation of the source function varies according to parameterization. The value of this function is presented in Fig. 9, row 4. All dust emission schemes utilize the
*EROD* field (referred to as the source strength $S$ in previous sections) to describe the availability of erodible soil in each grid cell. In the GOCART-WRF scheme, layers representing the fixed fractions of sand (50%), silt (25%), and clay (25%) are multiplied by the topographically-derived source function, $S$, (Eq. (4)) which ranges from 0 to 1. Since sand is excluded from the size fractions eligible for lofting, the sum of the fractions effectively varies from 0–0.5, halving the effective emissions. The AFWA scheme treats the dust emission flux as the physics-based flux times the *EROD* factor, which varies from 0 to 1. In
areas where $S$ is low, this may result in low emissions for the AFWA scheme compared to the UoC scheme. The UoC scheme uses the *EROD* factor as a binary dust source mask (i.e., if $EROD > 0$, the physics-based flux is turned on; if $EROD \leq 0$, no dust emission is allowed). An additional factor of $(1 - c_f)^2$, however, is also implemented at this stage in the UoC scheme, and so we incorporate the factor as part of the overall source correction displayed in Fig. 9, row 4. We see that the UoC source function is nonzero over a spatial domain much larger than the region emissions originate from. Therefore, our first hypothesis
above, that the binary source function was causing the limited emissions area in the UoC results is rejected.





The three models all go on to subsequently use the fluxes in Fig. 9 row 3, combined with the source terms in Fig. 9, row 4 to calculate the dust fluxes in Fig. 7, row 2. The GOCART-WRF and AFWA schemes amount to simple multiplications of the source terms and theoretical fluxes, with different methods for handling the parent soil particle size distribution and a small additional correction factor ($\beta$) in AFWA. The UoC conversion, with its consideration of soil makeup (Eq. (21)) and

bombardment efficiency, is quite different and more complex. Line by line comparison is not possible through these steps, but we note that the dust emissions in Fig. 7, row 2 are much higher in UoC than in AFWA for the (limited) locations having the same threshold friction velocity and source strength. For the purposes of explaining the limited spatial extent of the UoC emissions, the series of steps converting between saltation and dust emission in UoC favor higher dust emission, and thus are not the cause of limited emissions extent in UoC.

We conclude from this analysis that the primary cause of the differences in dust emissions between the AFWA and UoC schemes is the combined effect of multiple related terms. Emissions in western Iraq are restricted both by the surface roughness correction (Eqs. (18) and (19)) and the vegetation correction, $1 - c_f$, which is applied twice within the UoC scheme as dust flux is calculated from theoretical saltation flux. These effects all ultimately trace back to the vegetation fraction $c_f$. Through these parameterizations, the effect of small amounts of vegetation, which are apparently indicated in western Iraq within the

source dataset for $c_f$, are dominant in decreasing the erodibility of western Iraq and effectively shutting down emissions there.

The finding that the vegetation layer is essentially controlling the spatial extent of dust emissions in the UoC scheme highlights an important fact – the models are highly sensitive to terrain condition data inputs which are determined from notoriously sparse datasets. Though in this case, the AFWA scheme appears to produce dust emissions over a spatial domain in better agreement with observations, it would be challenging to conclude that this was related to superior model physics. Instead, the

primary cause of the UoC scheme's disagreement with observations appears to be spurious detection of vegetation coverage in western Iraq in the forcing dataset combined with a parameterization that permits vegetation coverage to excessively impact dust emissions. It is likely, though not investigated in this work, that changes in soil grain size data, which originate from similarly sparse datasets with limited validation, will have similarly large impacts.

Aside from improving vegetation coverage or soil composition data, we note that several tuning parameters are available

which could be used to attempt to better match behavior between models or better match model behavior to observations. The UoC model is particularly sensitive to the soil plastic pressure, and this variable is set to a constant for the entire model domain. Tuning this variable can result in matching the dust emissions of select regions, but not across the entire model domain, suggesting this parameter should be dependent on soil type.

## 8   Conclusions

The AFWA dust emission scheme for WRF-Chem is fully-documented in the literature for the first time here. This emission scheme represents a substantial advance in the physical realism of dust emission modeling over the GOCART-WRF emission scheme. Key improvements to model physics permit saltation flux, caused by aerodynamic entrainment, to be modeled separately from dust emission, largely caused by bombardment and disaggregation processes. Output from the model in a test



case is shown to broadly match the spatial distribution and intensity of dust emissions during a wintertime Shamal event in southwest Asia.

Analysis of the code and documentation available for the other dust emission schemes highlights several discrepancies between documentation and code implementation, as well as several changes in code implementation across WRF-Chem

versions that had not previously been documented. In particular, a recently corrected error in the implementation of the UoC scheme (see section 3.3.3) may have resulted in emissions from the implementation present in WRF-Chem versions obtained before the January 2018 bug fix release that were approximately an order of magnitude lower than would be expected from the parameterization that should have been included.

Comparing the parameterization approach of the AFWA scheme to the UoC scheme, as implemented in WRF-Chem 3.8.1,

highlights that the two models are similar in many ways. Though the physics included in the UoC dust emission scheme are potentially more physically complete, the AFWA model may have an advantage in mesoscale development due to its lower sensitivity to sparse and challenging to obtain soil and vegetation data. The most important future opportunities for improving both AFWA and UoC schemes appear to be related to the fixed input data on terrain properties. First and foremost, both schemes would benefit greatly from replacing the soil particle size distribution dataset and erodibility function with better

observational data. UoC would also benefit from improved soil and vegetation coverage data and from a function to make soil plastic pressure tied to soil type or particle size distribution. A focus on collecting and synthesizing such wide-ranging data on Earth surface characteristics will require a substantial, coordinated community effort.

*Code availability.* The GOCART model, as described in this document is embedded in WRF-Chem version 4.0, which is available to the community at http://www2.mmm.ucar.edu/wrf/users/. The three options for dust emission scenarios are coded as dust_opt=01 – "GOCART-

WRF", dust_opt=03 – "AFWA", and dust_opt=04 – "UoC".

**Appendix A: Variable list**

*Author contributions.* LeGrand and Creighton developed the AFWA dust emission scheme. Cetola supervised project execution of the AFWA scheme code development. Creighton and Peckham implemented the AFWA scheme code into the WRF-Chem framework. Letcher conducted and post-processed the WRF-Chem case study simulations. LeGrand, Polashenksi, and Letcher analyzed data and primarily wrote

the manuscript. All co-authors critically reviewed the manuscript.

*Competing interests.* The authors declare that they have no conflict of interest.





*Acknowledgements.* Funding support for this project was provided by the U.S. Army Terrestrial Environmental Modeling & Intelligence System (ARTEMIS) applied science research program sponsored by the Assistant Secretary of the Army for Acquisition, Logistics, and Technology (ASA-ALT). Access to CALIOP and MODIS data was provided by the NASA Earth Data Portal at search.earthdata.nasa.gov.



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





**Figure 1.** Schematic diagram of AFWA dust emission scheme and required inputs. The black diamond marker indicates that the parameter varies spatially and temporally. The black circle marker indicates that the parameter varies spatially, and the hollow diamond marker indicates the term is related to a particle size bin. See comprehensive variable list in Appendix A for variable definitions.

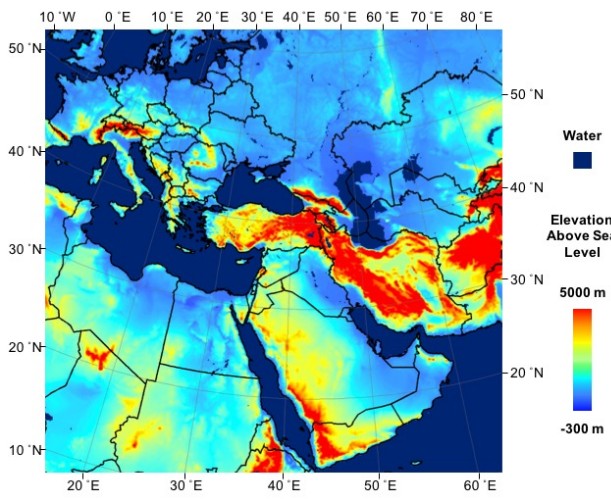

**Figure 2.** Domain map for the WRF-Chem simulations with color shading showing the waterbodies and elevation as indicated by the colorbar. The region of dust emissions we focus on is just right of center in the Syrian desert on both sides of the Iraq–Syrian border.





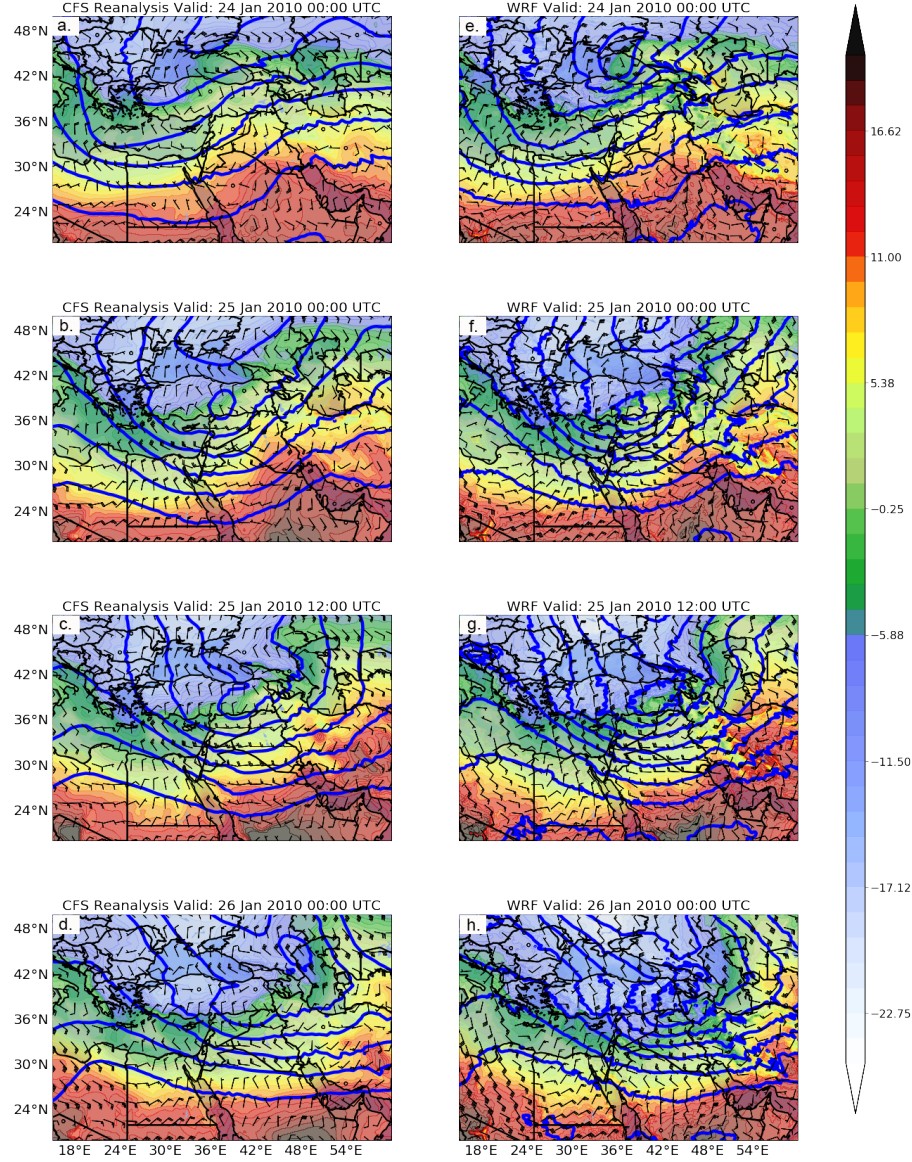

**Figure 3.** The synoptic environment during the time period surrounding the dust emission event. Blue lines represent 700hPa geopotential height, shading represents 850hPa temperature, and vectors represent 925hPa winds. Column at left (a–d) shows independent CFS reanalysis data. Column at right (e–h) shows WRF-Chem modeled conditions.



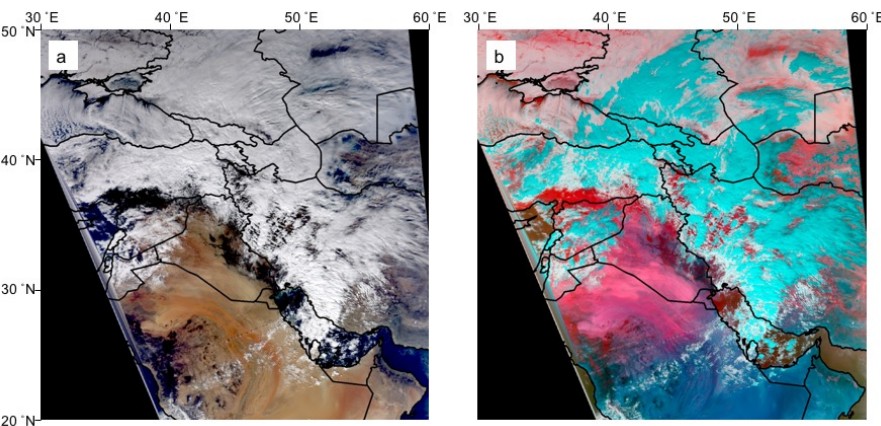

**Figure 4.** Observations of the lofted dust plume collected on 1000 UTC 25 January 2010 by the MODIS sensor including (a) true color composite and (b) dust-enhanced image produced using the Miller (2003) algorithm, where lofted dust appears pink, landscapes have blue and green hues, and water and steep terrain are red.





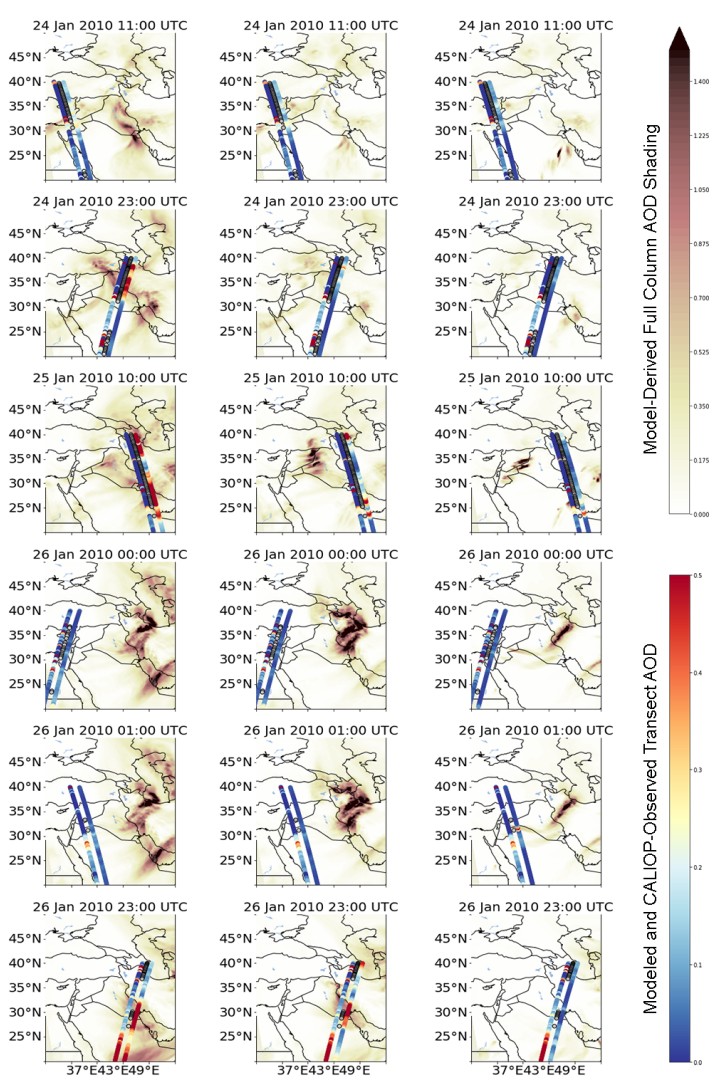

**Figure 5.** Shaded maps of modeled AOD for the GOCART-WRF (left column), AFWA (center column), and UoC (right column) scheme. Timestamps are indicated at the top of each image and are the same across the rows. Transects of AOD are also placed as overlays, with three adjacent transects representing observed AOD from the CALIOP data (left), locations along the transect where CALIOP observations are heavily impacted by cloud cover and retrieval does not represent full column AOD (center), and modeled full-column AOD along the transect (right).





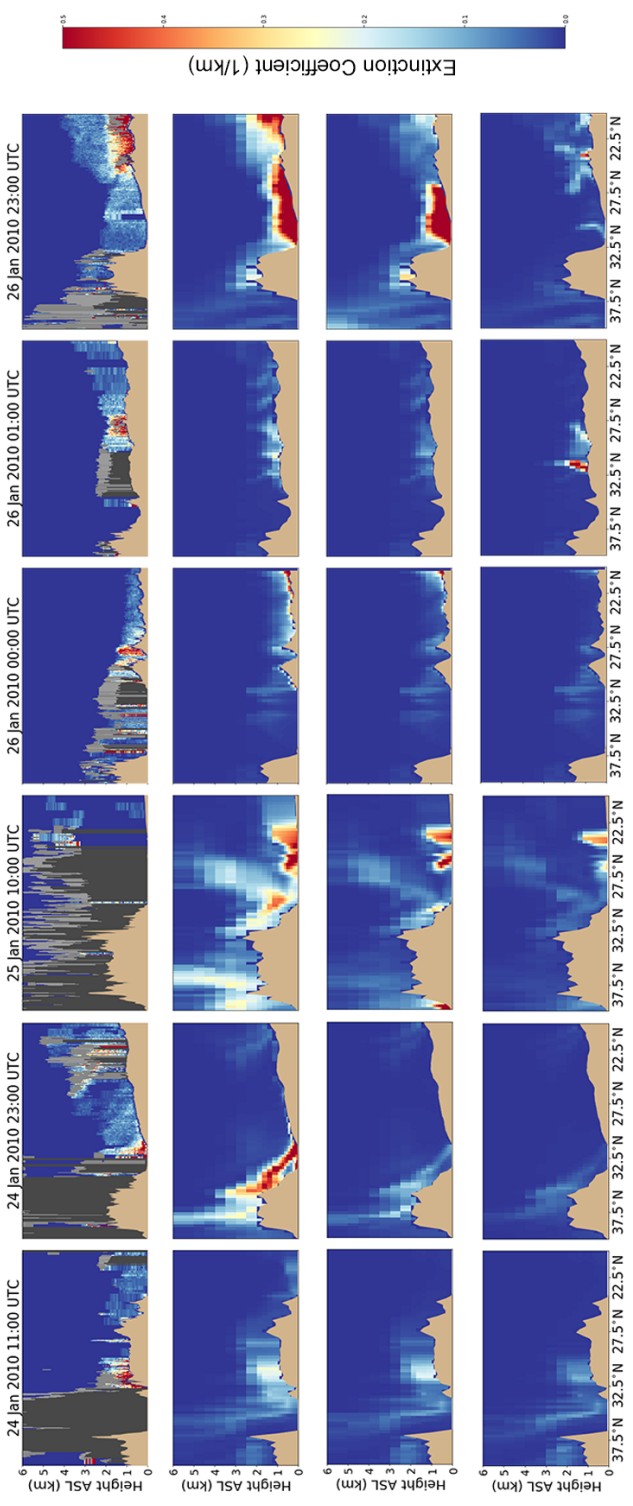

**Figure 6.** Aerosol extinction profiles at 550nm along the CALIOP transects for each of the six overpasses shown as integrated AOD in Fig. 5. Row 1 represents observed CALIOP data with light gray indicating clouds, and dark gray indicating areas beneath clouds where no data is available. Remaining three rows represent modeled data with GOCART-WRF in Row 2, AFWA in Row 3, and UoC in Row 4.





**Figure 7.** Modeled dust emissions from the GOCART-WRF (left column), AFWA (center column), UoC (right column) schemes. Time is indicated in the header, increasing from top to bottom. Note that times provided differ from Fig. 5 and 6. Here we provide snapshots at the times of the three CALIOP transect collections during the emission event and at three other times selected to show the evolution of the event. Emissions during the remainder of the time period represented by Fig. 5 and 6 are minimal.



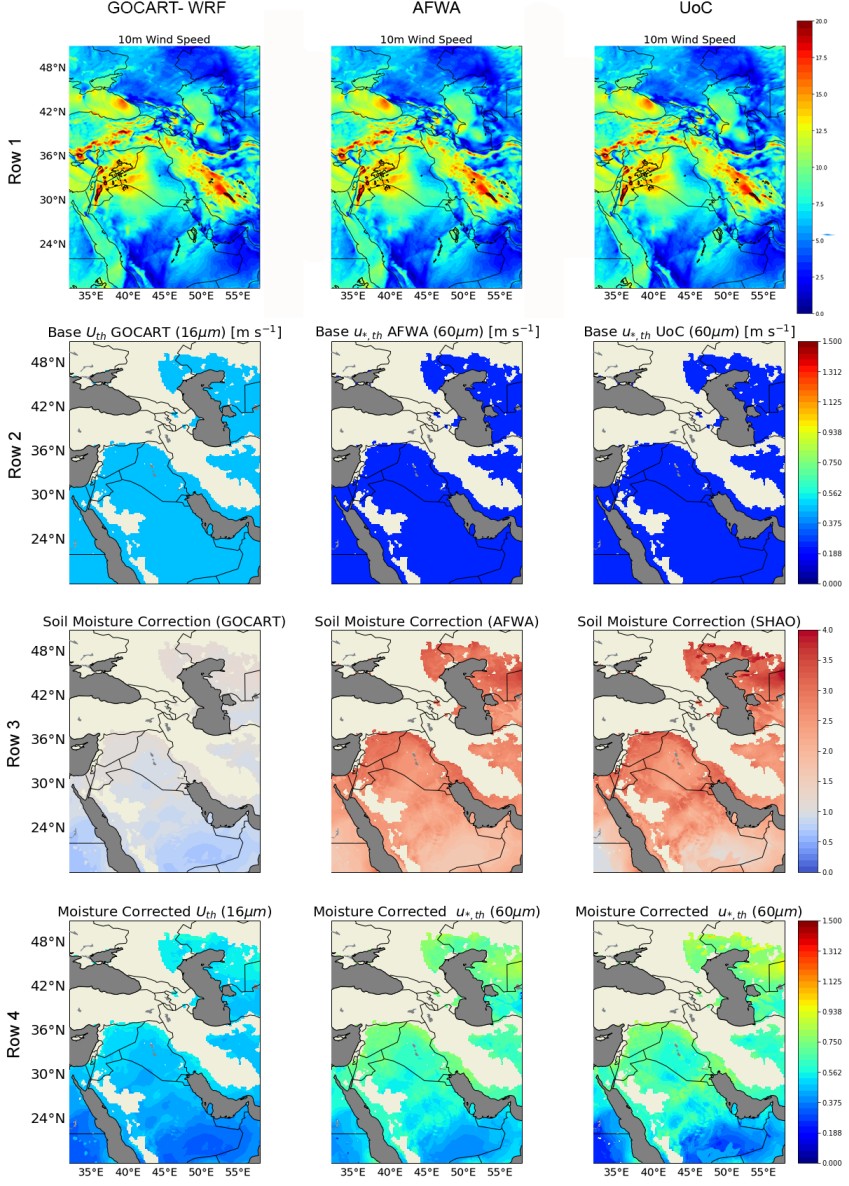

**Figure 8.** Values of intermediate variables used in the calculation of dust emissions by the three different emissions schemes, with the GOCART-WRF scheme in the left column, the AFWA scheme in the center column, and the UoC scheme in the right column. All images reflect model state at 1100 UTC on 25 January 2010. Wind speed is represented in row 1 (equivalent in all models), the theoretical dry soil threshold for saltation of grains having diameter 60 μm (16 μm for GOCART-WRF) is shown in row 2, the soil moisture correction factor applied is shown in row 3, and the moisture corrected threshold for saltation is shown in row 4. Areas of dark gray are water bodies, and areas void of color in rows 2–4 are areas masked out for vegetation in the source strength function.




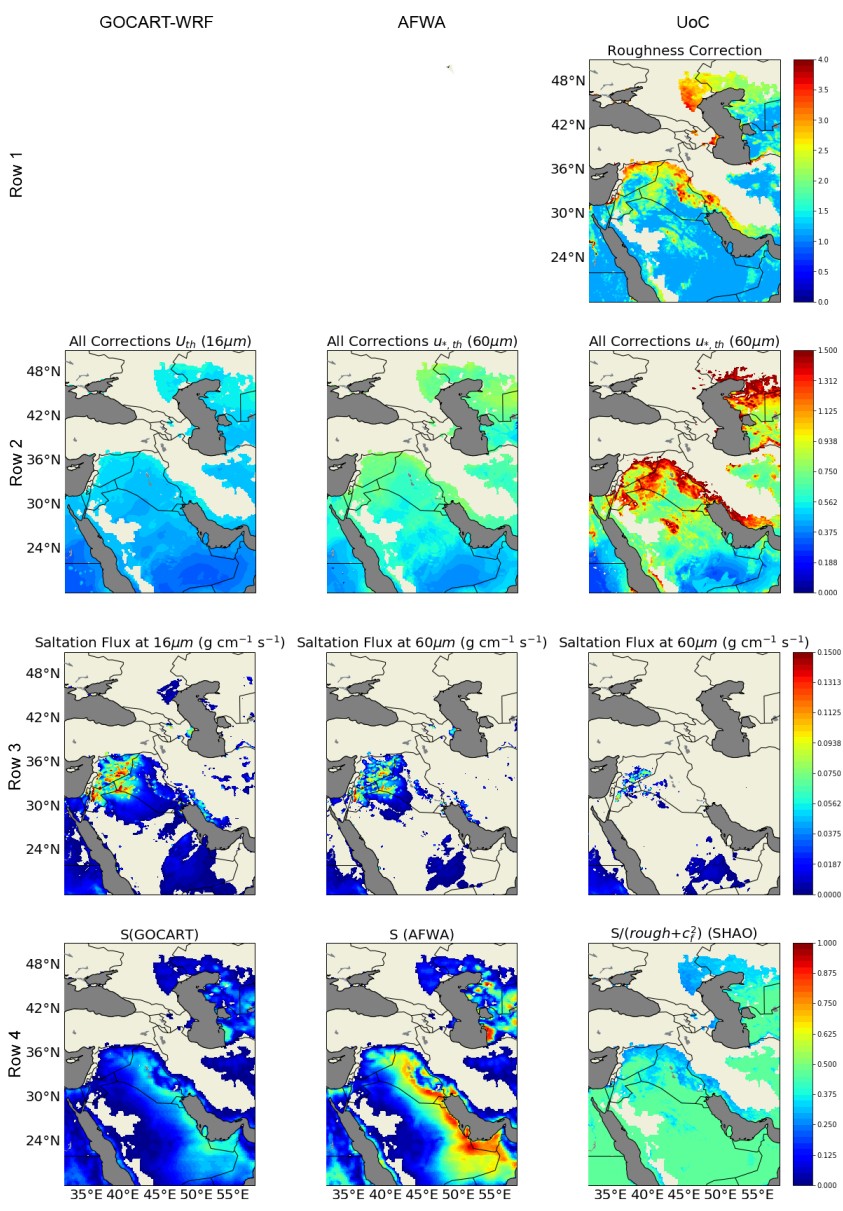

**Figure 9.** Building from Fig. 8, additional values of intermediate variables used in the calculation of dust emissions by the three different emissions schemes, with the GOCART-WRF scheme in the left column, the AFWA scheme in the center column, and the UoC scheme in the right column. All images reflect model state at 1100 UTC on 25 January 2010. The surface roughness correction, which only exists in the UoC scheme is presented in row 1, the threshold wind velocity for saltation after all corrections have been applied is shown in row 2, the saltation flux of 60 µm particles is shown in row 3 (dust emission flux for 16 µm particles in GOCART-WRF), and the source strength function is shown in row 4. Figure 7 Row 2 can be thought of as the next step in the calculation, and could also logically be considered here as if it were row 5.



**Table 1.** Saltation particle size bins and their associated attributes presented here in μm, but handled in units of cm within the model.

| Saltation Size Bin ($p$) | Lower Bound Diameter (μm) | Upper Bound Diameter (μm) | Effective Diameter ($D_{s,p}$) (μm) | Soil Separate Class | Soil Separate Class Mass Fraction ($s_{frac}$) | Particle Density ($\rho_p$) (g cm$^{-3}$) |
|---|---|---|---|---|---|---|
| 1 | 0.2 | 2 | 1.42 | Clay | 1 | 2.50 |
| 2 | 2 | 14 | 8 | Silt | 0.25 | 2.65 |
| 3 | 14 | 26 | 20 | Silt | 0.25 | 2.65 |
| 4 | 26 | 38 | 32 | Silt | 0.25 | 2.65 |
| 5 | 38 | 50 | 44 | Silt | 0.25 | 2.65 |
| 6 | 50 | 90 | 70 | Sand | 0.0205 | 2.65 |
| 7 | 90 | 170 | 130 | Sand | 0.0410 | 2.65 |
| 8 | 170 | 240 | 200 | Sand | 0.0359 | 2.65 |
| 9 | 240 | 1000 | 620 | Sand | 0.3897 | 2.65 |
| 10 | 1000 | 2000 | 1500 | Sand | 0.5128 | 2.65 |





**Table 2.** Dust particle size bins and their associated attributes, presented here in μm, but handled in units of cm within the model.

| Dust Size Bin ($p$) | Lower Bound Diameter (μm) | Upper Bound Diameter (μm) | Effective Diameter ($D_{d,p}$) (μm) | Particle Density ($\rho_p$) (g cm$^{-3}$) |
|---|---|---|---|---|
| 1 | 0.2 | 2 | 1.46 | 2.50 |
| 2 | 2 | 3.6 | 2.8 | 2.65 |
| 3 | 3.6 | 6 | 4.8 | 2.65 |
| 4 | 6 | 12 | 9 | 2.65 |
| 5 | 12 | 20 | 16 | 2.65 |





**Table 3.** Optional tuning parameters and binary configuration flags ($A$).

| Configuration Parameter | Purpose | Valid Values |
|---|---|---|
| $c_{ustune}$ | Threshold friction velocity tuning factor. | $0.0 \leq c_{ustune}$ |
| $c_{smtune}$ | Soil moisture tuning factor. Note, this parameter only affects soil moisture values as they are used in the correction function $f(\theta)$. Soil moisture values throughout the rest of the WRF-Chem model are not affected. | $0.0 \leq c_{smtune}$ |
| $c_{\gamma}$ | Exponential tuning factor applied to the preferential dust source term, $S$. Setting $c_{\gamma} > 1.0$ will decrease the spatial footprint of the dust sources when using the original WRF-Chem $S$ described by Ginoux et al. (2001) since these values are less than 1.0. | Any float |
| $c_{\alpha}$ | Bulk vertical dust emission flux tuning factor. | $0.0 \leq c_{\alpha}$ |
| $A_{DSR}$ | Flag to utilize an alternate, user provided preferential dust source strength term. | 1 to activate; 0 otherwise |
| $A_{VEG}$ | Flag to apply a user provided vegetation mask to the $S$ parameter. | 1 to activate; 0 otherwise |
| $A_{SOILS}$ | Flag to utilize alternate, user provided sand and clay mass fraction datasets. | 1 to activate; 0 otherwise |
| $A_{SMOIS}$ | Flag to utilize an alternate form of the $f(\theta)$ calculation as described by Hunt et al. (2014). Use of this modification removes the need for the $\theta_v$ to $\theta_g$ conversion. | 1 to activate; 0 otherwise |





**Table 4.** Alternate saltation particle size bins and their associated attributes as implemented in WRF-Chem versions 3.4–4.0.

| Saltation Size Bin ($p$) | Effective Diameter ($D_{s,p}$) (μm) | Soil Separate Class | Soil Separate Class Mass Fraction ($s_{frac}$) | Particle Density ($\rho_p$) (g cm$^{-3}$) |
|---|---|---|---|---|
| 1 | 1.42 | Clay | 1 | 2.50 |
| 2 | 2.74 | Silt | 0.2 | 2.65 |
| 3 | 5.26 | Silt | 0.2 | 2.65 |
| 4 | 10 | Silt | 0.2 | 2.65 |
| 5 | 19 | Silt | 0.2 | 2.65 |
| 6 | 36.2 | Silt | 0.2 | 2.65 |
| 7 | 69 | Sand | 0.333 | 2.65 |
| 8 | 131 | Sand | 0.333 | 2.65 |
| 9 | 250 | Sand | 0.333 | 2.65 |





**Table 5.** WRF-Chem physics and chemistry parameterizations.

| Parameterization | Scheme | Namelist Variable | Option |
|---|---|---|---|
| Cumulus | Kain-Fritsch (Kain, 2004) | cu_physics | 1 |
| Surface Model | Noah (Tewari et al., 2004) | sf_surface_physics | 2 |
| Surface Layer | MM5 (Beljaars, 1994) | sf_sfclay_physics | 1 |
| Boundary Layer | MYNN 2.5 (Nakanishi and Niino, 2006) | bl_pbl_physics | 5 |
| Radiation (SW & LW) | RRTMG (Iacono et al., 2008) | ra_sw(lw)_physics | 4 |
| Microphysics | Thompson (Thompson et al., 2008) | mp_physics | 8 |
| Chemistry | GOCART Simple / No ozone chemistry | chem_opt | 300 |
| Background Emissions | GOCART Simple | emiss_opt | 6 |
| Aerosol Optics | Maxwell Approximation | aer_op_opt | 2 |



**Table A1.** Variable list.

| Variable | Name | Value | dust_opt | Equations |
|---|---|---|---|---|
| $A$ | Dimensionless Constant | 6.5 | 1 | 2 |
| $A_n$ | Dimensionless Constant | 0.0123 | 4 | 17, 36 |
| $a$ | Dimensional Constant | 1331 cm$^{-x}$ | 1, 3 | 5 |
| $b$ | Dimensionless Constant | 6.5 | 1, 3 | 5 |
| $C$ | Dimensional Constant | $10^{-9}$ kg s$^2$ m$^{-5}$ | 1 | 1 |
| $C_{mb}$ | Dimensionless Constant | 1 | 3 | 20 |
| $c_f$ | Vegetation Fraction | Constant field | 4 | 19, 20, 30, 33 |
| $c_s$ | Soil Clay Content Mass Fraction | Constant field | 3 | 8, 9 |
| $c_{smtune}$ | Soil Moisture Tuning Constant | User set | 3 | Fig. 1 |
| $c_{ustune}$ | Friction Velocity Tuning Constant | User set | 3 | Fig. 1 |
| $c_v$ | Dimensionless Constant | 12.62 x $10^{-4}$ cm | 4 | 15 |
| $c_y$ | Dimensionless Constant | 0.00001 | 4 | 23, 27, 28 |
| $c_\alpha$ | Source Strength Tuning Constant | User set | 3 | Fig. 1 |
| $c_\gamma$ | Dust Emission Flux Tuning Constant | User set | 3 | Fig. 1 |
| $D_{d,p}$ | Particle Diameter of Dust Bin Size $p$ | Variable | 3 | 15 |
| $D_{d,p\_max}$ | Max Particle Diameter of Dust Bin Size $p$ | Variable | 3 | 15 |
| $D_{d,p\_min}$ | Min Particle Diameter of Dust Bin Size $p$ | Variable | 3 | 15 |
| $\bar{D}_m$ | Dust particle Mass Median Diameter | 3.4 x $10^{-4}$ cm | 3 | 15 |
| $D_p$ | Particle Diameter, Bin Size $p$ | Variable | 1, 4 | 1, 2, 3, 5, 17 |
| $D_{s,p}$ | Particle Diameter of Saltation Bin Size $p$ | Variable | 3 | 5, 6, 10, 11, 12, 13 |
| $d$ | Particle Diameter | Variable | 4 | 17, 20, 21, 22, 25, 36 |
| $d_i$ | Dust Particle Diameter | Variable | 4 | 24 |
| $d_s$ | Saltation Particle Diameter | Variable | 4 | 24 |
| d__ | Distributions of Particle Property __ | Variable field | 3 | |
| dM | Particle Mass Distribution Fraction | Variable field | 3 | 11 |
| $dS_{SFC}$ | Particle Basal Surface Coverage Fraction | Variable field | 3 | 11, 12 |
| $dS_{rel}$ | Relative Weighting Factors for Particle Size Bins | Variable field | 3 | 12, 13 |
| $dV_{d,p}$ | Normalized Volume Distribution for Dust Bin $p$ | Variable field | 3 | 15 |
| $F$ | Dust Emission Flux | Variable field | 4 | 24, 27, 28, 30 |
| $F_B$ | Bulk Dust Emission Flux | Variable field | 3 | 14 |
| $F_{d,p}$ | Dust Emission Flux in Dust Bin Size $p$ | Variable field | 3 | 16 |
| $F_p$ | Dust Emission Flux Bin Size $p$ | Variable field | 1 | 1 |
| $F_{total}$ | Dust Emission Flux | Variable field | 4 | 32 |
| $f$ | Moisture Correction Function | Variable field | 3, 4 | 6, 7 |
| $G$ | Streamwise Horizontal Saltation Flux | Variable field | 3 | 13, 14 |

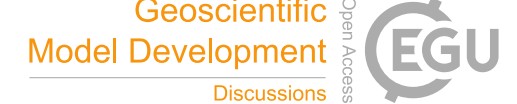

**Table B1.** Variable list continued.

| Variable | Name | Value | dust_opt | Equations |
|---|---|---|---|---|
| $g$ | Gravitational Acceleration Constant | 9.81 m s$^{-2}$ | 1, 3, 4 | 2, 5, 10, 17, 20, 24, 27, 28, 33 |
| $H$ | Saltation Flux in Bin Size $p$ | Variable field | 3 | 10, 13 |
| $k_1$ | Aggregate Breakup Constant | 1.0 | 4 | 23 |
| $m$ | Mass of a Particle | Variable | 4 | 24 |
| $N_{SFC}$ | Total Basal Surface Area of Soil Bed | Variable field | 3 | 12 |
| $N_v$ | Total Normalized Emitted Dust Volume | Variable field | 3 | 15 |
| $p$ | Particle Availability Term Variable field | Variable field | 4 | 21, 22 |
| $p_m$ | Minimally Disturbed Particle Size Distribution | Variable field | 4 | 22 |
| $p_f$ | Fully Disturbed Particle Size Distribution | Variable field | 4 | 22 |
| $Q$ | Source Corrected Saltation Flux | Variable field | 4 | 21, 29 |
| $Q_{TOTAL}$ | Particle Size Bin Integrated Saltation Flux | Variable field | 4 | 28, 29 |
| $q$ | Theoretical Saltation Flux | Variable field | 4 | 20, 27, 33 |
| $r$ | Roughness Correction Factor | Variable field | 4 | 18 |
| $S$ | Dust Source Strength Function | Variable field | 1, 3, 4 | 1, 4, 14 |
| $S_b$ | Binary Dust Source Function | Variable field | 4 | 21 |
| $s_{frac}$ | Soil Separate Class Mass Fraction | Variable field | 3 | 11 |
| $s_p$ | Soil Surface Mass Fraction, Bin Size $p$ | Variable field | 1 | 1 |
| $U$ | 10m Wind Speed | Variable field | 1 | 1 |
| $U_p$ | Particle Impact Velocity | Variable | 4 | 25, 36 |
| $U_t$ | Threshold 10m Wind Speed | Variable field | 1 | 1, 2, 3 |
| $u_*$ | Wind Friction Velocity | Variable field | 3, 4 | 10, 20, 23, 24, 26, 27, 28, 33, 34, 35 |
| $u_{*t}$ | Threshold Wind Friction Velocity | Variable field | 3, 4 | 5, 17, 20, 23, 33, 34, 35, 36 |
| $x$ | Dimensionless Constant | 1.56 | 1, 3 | 5 |
| $x_f$ | Frontal Area Index | Variable field | 4 | 18, 19 |
| $z_{max}$ | Highest Topographic Point in 10° x 10° Area | Constant field | 1, 3, 4 | 4 |
| $z_{min}$ | Lowest Topographic Point in 10° x 10° Area | Constant field | 1, 3, 4 | 4 |
| $z_i$ | Topographic Elevation, cell $i$ | Constant field | 1, 3, 4 | 4 |
| $\alpha_i$ | Incidence Angle of Collisions | 15 ° | 4 | 25, 36 |
| $\beta$ | Soil Crusting Factor | Constant field | 3 | 14 |
| $\beta_v$ | Bombardment Factor | Variable | 4 | 25, 36 |
| $\gamma$ | Aggregation Strength Parameter | Variable field | 4 | 22, 23, 24, 27 |
| $\gamma_c$ | Dimensional Constant | 1.65x10$^{-4}$ kg s$^{-2}$ | 4 | 17 |

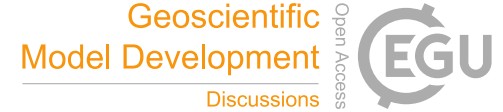



**Table C1.** Variable list continued.

| Variable | Name | Value | dust_opt | Equations |
|---|---|---|---|---|
| $\eta_{c,i}$ | Soil Fraction Available for Disaggregation, Bin $i$ | Variable field | 4 | 24 |
| $\eta_{f,i}$ | Fully Disturbed Dust Fraction, Bin $i$ | Variable field | 4 | 24, 27 |
| $\eta_{m,i}$ | Minimally Disturbed Dust Fraction, Bin $i$ | Variable field | 4 | 24, 28 |
| $\theta$ | Soil Moisture | | | |
| $\theta_g$ | Gravimetric Soil Moisture Fraction | Variable field | 3, 4 | 7, 9 |
| $\theta_g'$ | Fraction of Moisture w/o Effect on Capillary Forces | Variable field | 3, 4 | 7, 8 |
| $\theta_{gc}$ | $\theta_g$ multiplied by constant tuning factor | Variable field | 3 | Fig. 1 |
| $\theta_s$ | Moisture Fraction, % Saturation | Variable field | 1 | 1, 2. 3 |
| $\theta_v$ | Volumetric Soil Moisture Fraction | Variable field | 3, 4 | 9 |
| $\kappa_{d,p}$ | Size Distribution Weighting Factor, Dust Size Bin $p$ | Variable | 3 | 15 |
| $\lambda$ | Crack Propagation Length | 12.0 x $10^{-4}$ cm | 3 | 15 |
| $\pi$ | Pi | 3.14159 | 4 | 25, 36 |
| $\phi$ | Soil Porosity | Constant field | 3 | 9 |
| $\rho_a$ | Air Density | Variable field | 1, 3, 4 | 2, 5, 10, 20, 33 |
| $\rho_b$ | Constant Bulk Density of the Soil | 1000 kg m$^{-3}$ | 4 | 24, 26 |
| $\rho_p$ | Particle Density, Size Bin $p$ | 2.5–2.65 g cm$^{-3}$ | 1, 3, 4 | 2, 17, 36 |
| $\rho_{s,p}$ | Particle Density, Saltation, Size Bin $p$ | 2.5–2.65 g cm$^{-3}$ | 1, 3, 4 | 5, 11 |
| $\rho_w$ | Water Density | 1.0 g cm$^{-3}$ | 1, 3, 4 | 2, 9, 17 |
| $\varrho$ | Soil Plastic Pressure | 30000 N m$^{-2}$ | 4 | 25, 26, 36 |
| $\sigma_m$ | Revised Bombardment Efficiency | Variable Field | 4 | 26, 27, 28 |
| $\sigma_p$ | Ratio of Particle Density to Air Density | Constant | 4 | 17, 24, 27, 36 |
| $\sigma_s$ | Geometric Standard Deviation | 3.0 | 3 | 15 |
| $\Omega$ | Bombardment Efficiency | Variable Field | 4 | 24, 36 |