# Peer review of "The AFWA Dust Emissions Scheme for the GOCART Aerosol Model in WRF-Chem"

_Geoscientific Model Development, 2018_

## Short Comment (SC1) · 15 Aug 2018

Dear authors,

In my role as Executive editor of GMD, I would like to bring to your attention our Editorial version 1.1:

http://www.geosci-model-dev.net/8/3487/2015/gmd-8-3487-2015.html

This highlights some requirements of papers published in GMD, which is also available on the GMD website in the 'Manuscript Types' section:

http://www.geoscientific-model-development.net/submission/manuscript_types.html

In particular, please note that for your paper, the following requirements have not been

met in the Discussions paper:

- "The main paper must give the model name and version number (or other unique identifier) in the title."

- "All papers must include a section, at the end of the paper, entitled 'Code availability'. Here, either instructions for obtaining the code, or the reasons why the code is not available should be clearly stated. It is preferred for the code to be uploaded as a supplement or to be made available at a data repository with an associated DOI (digital object identifier) for the exact model version described in the paper. Alternatively, for established models, there may be an existing means of accessing the code through a particular system. In this case, there must exist a means of permanently accessing the precise model version described in the paper. In some cases, authors may prefer to put models on their own website, or to act as a point of contact for obtaining the code. Given the impermanence of websites and email addresses, this is not encouraged, and authors should consider improving the availability with a more permanent arrangement. After the paper is accepted the model archive should be updated to include a link to the GMD paper."

Thus, at least add to the title the version number of the WRF version which includes the exact dust emission schemes discussed here. It would even be better to provide version numbers for the GOCART model and / or the dust emission scheme.
Regarding the Code availability, please ensure that the exact version described here is available to the reader. Additional, add the information how to access the code.

Yours,

Astrid Kerkweg
* * *

---

## Referee Comment (RC1) · Anonymous Referee #1 · 20 Aug 2018

This manuscript provides a full documentation of the AFWA dust emission scheme in WRF-Chem, and the differences between dust emission produced by the three available dust emission schemes are also discussed. The manuscript is well-written and clearly presented.

Suggestion on the manuscript.

The authors should apply more observational data to show the difference between the results produced by the three dust emission schemes? Such as the daily AOD map from MODIS, or the surface AOD observations from AERONET program? It is hard to conclude with only the transects of CALIPSO extinction coefficient.

---

## Referee Comment (RC2) · Anonymous Referee #2 · 3 Sep 2018

The authors describe and compare the current dust emission options in WRF-Chem and discuss similarities and differences between the different options. The objective of the paper is to document the AFWA-dust emission module in WRF, but strong emphasis is given also on the GOCART and UoC dust modules with the goal to compare the implementations and document so-far undocumented aspects. While this is useful, it does not seem to have happened with interaction/consultation of the persons responsible for the implementations, which is - at least - surprising and which might have helped to clarify certain aspects.

The paper is overall well written and organized. However, there are several shortcomings/incorrect statements, in particular regarding the description of the UoC implementation. I also see some problems regarding the terminology and code versions used

for the simulations. I recommend revision of the manuscript, considering the following comments:

GOCART-WRF Implementation:

- The authors discuss the change of an expression for the saltation threshold in the GOCART-WRF implementation from one for wind velocity to one for friction velocity. It is important to note here that both equations for threshold velocity (Eqs. 2 and 5) were originally expressions for threshold friction velocity, only the coefficient A in Eq. 2 was adapted, supposedly to mimic a wind speed rather than friction velocity. The deficits discussed in section 3.1.2 could therefore be easily overcome by either doing a similar empirical adjustment or by using one of the stability functions to convert between u* and u readily available from the surface layer physics in WRF. The authors further discuss that the use of such a threshold friction velocity would be "physically invalid" (P9 L16), because it is designed to represent the initiation of saltation (P8 L26) while saltation is not explicitly represented in the GOCART-WRF scheme. This argument does not hold, because the merging of saltation and dust emission to one empirical relationship in the parameterization does not contradict the assumption that dust emission is initiated by saltation. This is also stated by the authors themselves (P5 - L15-19): "The impacts of saltation bombardment processes on mobilization are not necessarily omitted - rather they are internalized in the relationship between wind speed and emissions". For this reason, I suggest still to highlight the issue of comparing u* with u in the current implementation also mentioning that a correction like it was done before could easily be added, but to remove the discussion about the unphysical use of the equation (in an empirical parameterization) at the end of Section 3.1.2, the purpose of which seems to be mainly to motivate the introduction of the AFWA module. This is unnecessary. The formulations in this motivating paragraph, i.e. P9 L14-L22, to me also seem to be too strong statements in terms of the novelty of the implementation keeping in mind that it is not a new emission parameterization, but the incorporation of existing and well-known parameterizations in WRF. Apart of that, I recommend to add

references to Eqs. (2) [Bagnold, 1941; Ginoux et al. (2001)].

- P6 L15 The authors state that the impact of a soil moisture correction factor > 1 is small, because soils moisture does not normally assume such small values "in most numerical weather models". It would be more relevant here to discuss this in the framework of WRF which does seem to allow for such small values (P28 L23-24).

- P8 L7-9 The mismatch between predicted and observed threshold friction velocities for small particles in the Bagnold-parameterization is well-known and dates back to the mid/late 20th century. Iversen and White (1982) provided the next well-referenced parameterization for u*t including a minimium of u*t for particles of about 100 micrometers in diameter (Iversen and White, 1982 is also the basis for the MB95 expression used in the AFWA implementation), followed by Shao and Lu (2000), who put the expression on pure physics-based footing. Reference to a modeling study from 2003 does therefore not seem appropriate here.

- P8 L18-19 It is explained here that the coefficient in Equation 15 (0.129) is given as 0.0013 in the model due to rounding and unit conversion. However, checking the source code, I see a factor of 0.13 (L. 273 in module_gocart_dust.F and L.511 in module_gocart_dust_afwa.F, WRF-Chem V4.0). Please clarify.

AFWA implementation:

- P9 L26 The MB95 parameterization represents saltation bombardment only.

- Repetition of Eq. (5) seems unnecessary here.

- P9 L12 - See previous comment on the factor 0.0013

- Please add reference to Eq. (10)

UoC implementation:

- P14 L7 The namelist variable is called dust_schme and not dust_scheme.

- P14 L12 ["Both schemes simulate the physics of dust emission"] This is not correct. While the Shao schemes used in the UoC module are physics-based parameterizations, the AFWA module makes use of the Marticorena and Bergametti parameterization, which is semi-empirical. See also my later comments on "physics-based schemes" and the technical term "schemes" under "Terminology".

- P14 L15 Which dust emission bins are referred to here, the bins to calculate the emissions or the bins passed on to the WRF transport routines? The former are not the same between the UoC and AFWA modules and the latter are consistent with the GOCART-WRF and AFWA implementations only from WRF V3.8.1. Before that the UoC implementation was using different bins (see Flaounas et al., 2017)

- Note that while Eq. (17) might give similar output like Eq. (5), it is not empirical.

- P14 L25 The value of $1.65 \times 10^{-4}$ kg s$^{-2}$ is documented in Darmenova et al. (2009)

- P15 L18 I strongly recommend not to merge coefficients here, as this can give an equation a different appearance. Please list all coefficients separately for consistency with the original references.

- P15 L7-8 The UoC implementation uses the vegetation fraction provided by the WRF-model. This can easily and should be updated for case studies to obtain more accurate results. The specific vegetation product used is therefore not a feature of the UoC dust emission module, but of the parent WRF model.

- P16 L2-3 The statement here is unclear and misleading. Supply-limited saltation is not accounted for in either of the implementations in WRF. While the EROD function is meant to represent the availability of erodible sediment, it does by no means account for supply limitation in its physical meaning within the saltation process. Rather, it represents the "most probable locations of sediment" (Ginoux et al., 2001).

- P16 L7-9 This sentence is not clear to me.

- P16 L10 the variable dpsds is not calculated using Eq. (22). Eq. (22) gives the probability density function for airborne sediment particle-size distribution p_s(d) ("psds" in the code) (e.g. S11). Please modify Eq. (22) accordingly for consistency with S11. "dpsds" is the probability for each bin and follows according to the definition of probability density functions. There is therefore no need to introduce such an internal variable here.

- P16 L14 d is diameter, not bin.

- P16 L15-16 ["Limitations..."] This seems to be a general statement and not specific to the UoC implementation.

- P16 L19-20 ["prior to correction for soil moisture and ground cover"] This is not correct, the corrections are applied first.

- P17 L7 ["other tuning parameters"] While soil characteristics like the ones mentioned can be used to tune a model, they are not per se tuning parameters, but have a physical meaning.

- Eq. (25) I do not understand how the authors derived this equation. It is inconsistent with the one implemented in the UoC-S01 module. See also my comment further down on Section 3.3.2, Point 6. Apart of that, it needs to be $Q(d_s)$ rather than $q(d_s)$.

- Eq. (27) $Q(d_s)$ rather than $q(d_s)$

- P18 L16 The authors discuss here about a vegetation correction applied on both saltation and dust emission flux in the model and speculate that this correction "may be in error". The correction effectively reduces the surface area from which (a) sand particles and (b) dust particles can be emitted. Considering emission as a two-part process, application of the correction twice, i.e. for Q and F separately, is therefore plausible.

- P19 L6 The authors claim that "measurements of these soil characteristics are generally unavailable", referring to the use of soil particle-size distributions. This is surprising given that a complete set of parameters representing particle-size-distributions for the

12 USDA soil-texture classes is provided with the implementation. Availability is therefore not an issue and can be considered similar to that of other "difficult-to-obtain" soil-related parameters, e.g. porosity or clay fraction as used in the AFWA implementation.

- P19 L10-18 The description of how the soil particle-size distributions are obtained is not clear. The use of the FAO soil map is again, like vegetation cover, that provided by the WRF modeling framework and should not be considered as a feature of the implementation. The term "soil modes" is also misleading in the context of probability density functions, for which a "mode" has a statistical meaning. The soil parameters available in the UoC implementation are assigned to the 12 USDA soil texture classes for each of which particle-size distributions can be computed. Further, the particle-size distributions are calculated in the subroutine psd_create and not in the subroutine h_c. The latter determines the moisture correction of the threshold friction velocity. However, I believe that the names of individual subroutines should not be discussed here.

- In the original paper S04, c_y varies from 1 x 10ˆ-5 to 3 x 10ˆ-4. Note that exponential notation (1 x 10ˆ-5 rather than 1e-5) is preferable.

- Sec. 3.3.2, Point 2 - documented in Darmenova et al. (2009), see comment above

- Sec. 3.3.2, Point 3 - The roughness correction represents drag partition, while the application of (1-cf) correct for the area covered by vegetation. The factor is discussed in Darmenova et al. (2009).

- Sec. 3.3.2, Point 4 - The use of the Kawamura/White saltation flux equation is documented in Shao et al. (2011), in which also the Shao (2004) scheme is used.

- Sec. 3.3.2, Point 5 - See earlier comment on "soil modes"

- Sec. 3.3.2, Point 6 - This point is also incorrect. First, Eq. (25) is not the one implemented in the model. In the relevant subroutine (vhlys), it is stated clearly that the subroutine computes Eq. (8) from Lu and Shao (1999). Comparing the implementation with Eq. (8) in the original paper shows that the two are in perfect agreement. The supposed difference of a factor of 1/d mentioned by the authors disappears understanding that Eq. (8) in Lu and Shao (1999), gives V/b rather than V and that b is approximately equal to d as explained in Shao (2001). The reason why the Equation from Lu and Shao (1999) is implemented here is likely the fact that the new Equation in Shao (2001) is more complicated and subject to further testing as is discussed at length in Shao (2001). Second, Eq. (36) [also Eq. 36 in Shao, 2001] is also in perfect agreement with Eq. (8) in Lu and Shao (1999), which can easily be show using mathematical conversions and inserting beta, while the Equation given by the authors (their Eq. (25)) is incorrect.

- P21 L21 The Shao schemes available in the UoC module do not include aerodynamic (dust) entrainment. In Shao et al. (2001), Section 5 it is stated: "Here we are mainly concerned with the latter case" referring to saltation-based dust emission

- P21 L27 Eq. (7) in Shao (2004) does not represent sigma_p. Eq. (7) describes gamma (cf. Eq. (23) in the present paper).

Test case and comparison:

- P22 L13 The references given here belong to WRF-Chem, not to the dust emission schemes. I suggest moving them to an earlier position.

- If the UoC saltation flux bug fix was released in January 2018, this was well before submission of the manuscript. The version used for evaluation in this paper should therefore be the one with the bug corrected. There is no point in using a version that is known to be wrong and that is outdated. If the authors wish to test the effect of this bug fix on the results, they can do so in an appendix, but the version in the main text should be the version "as is", i.e. including the bug correction.

- In Section 3.2, an implementation error is mentioned for the AFWA implementation. It is not clear whether the version used in the comparison is the one with or without

the error correction. The same as mentioned in the previous comment for the UoC scheme applies here, too, with the only difference that the correction for the AFWA scheme does not seem to be included in the current release, but will be in a future version.

- P23 L20-21 ["The atmospheric dust observed..."] Please add reference, e.g. a figure, or give additional explanation.

- P24 L25-26 It is sufficient to give the color coding in the figure caption.

- P25 L28, P27 19 I suggest adding one or two more references for the "spurious dust lofting" in the GOCART-WRF implementation if available, keeping in mind that - if it depends on u*t vs. ut - this could be relatively easily fixed.

- P26 L9-10 The larger spatial extent in the results of the GOCART-WRF scheme are visible most of the time in Fig. 5, but not at 10 UTC on 25 Jan for which the MODIS data is shown in Fig. 4.

- P27 L24 (and relevant subsequent passages) The binary use of the EROD function cannot cause a reduced area of active dust emission in the UoC parameterization: dust emission is possible wherever EROD > 0, i.e. wherever dust emission is possible in the AFWA implementation.

- P27 L29 The version using the bug fix should be used here - see earlier comment.

- P28 L5 Is the threshold friction velocity meant with "soil threshold parameter"? In that case it would depend on particle size and not be a single value.

- P28 L16 The coefficients used in the soil moisture correction are not only different due to different units. Different sets of coefficients are also used for each of the 12 soil texture classes (Klose et al., 2014; based on Shao and Jung, 2000, unpublished manuscript)

- Fig. 8, If the same meteorology is used for all runs, it would be sufficient to show wind

speed only once.

- Fig. 9, All corrections - Why are there no values shown north-west of the Caspian Sea for the UoC implementation?

- P 29 L12-21 See previous comments on bug fix.

- P29 L32/Fig. 9 Please explain why S/(rough+cfˆ2) is plotted here.

Terminology:

- The terms scheme, parameterization, and model are used almost interchangeably here. This is problematic, in particular in the context of the GOCART, AFWA and UoC "schemes", which in my opinion are neither scheme nor parameterization nor model, but only the implementations of existing parameterizations/schemes in a model (which would be WRF-Chem in this case). I think it is important to use consistent terminology throughout the paper.

- The authors use the expression "emission mode" at several locations (e.g. P4 L3, P4 L15, P5 L20). I am not aware of any common use of this expression in the dust emission/aeolian community. I would therefore strongly recommend to abstrain from this expression. Most likely it is being confused with the modes of particle motion, which are, e.g., saltation, suspension, creep (Bagnold (1941), Shao (2008), Kok et al. (2012)). Please revise.

- P5 L16-18 The explicit separation of saltation and dust emission fluxes in a parameter-ization does not necessarily make it a physics-based parameterization. If the saltation flux and/or dust emission flux are represented by empirical relationships rather than basic physics, it will still be (semi-)empirical. The text should be modified accordingly.

Minor comments:

P1 L13 - particles rather than particulates

P2 L5 - Reference for GOCART model needed here, in particular the dust component

that is of relevance for this paper.

P2 L9 - "enabling their vertical movement" is not correct here speaking of dust emissions - Please revise, e.g. ""enabling dust transport in the atmosphere"

P2 L11 - As the present paper is concerned with dust emission, the addition of Ginoux et al. (2001) as a reference here would be appropriate.

P3 L9-10 - Implementation described in Darmenova et al. (2009)

P3 L22-23 - aerodynamic lift, saltation bombardment, and particle disaggregation are not forces, but processes. The half-sentence introducing those is misleading.

P9 L25 saltation bombardment

P9 L29 "effective particle size" rather than "effective aerosol size"

P22 L18-19 reference to NOAA/NCEP (2000) in parentheses

---

## Author Comment (AC1) · 1 Nov 2018

We have changed the title of the manuscript to include a version identifier: "The AFWA Dust Emissions Scheme for the GOCART Aerosol Model in WRF-Chem v3.8.1"

We have also updated the code availability section to include a more direct link, changed the WRF-Chem version number from the most recent release to the specific version used in this study, and listed key configuration file settings required to run the WRF-Chem model with the three dust emission schemes: "The code used in this study (WRF-Chem v3.8.1) is included in the chemistry package of the WRF model, currently available through http://www2.mmm.ucar.edu/wrf/users/download/get_sources.html. Users can select from the three dust emission schemes discussed by setting

[Figure]

*dust_opt=1* for GOCART-WRF, *dust_opt=3* for AFWA, or *dust_opt=4* for UoC in the namelist.input configuration file. If the UoC scheme is selected, the user must also choose one of the UoC sub-options by setting *dust_schme=1* for S01, *dust_schme=2* for S04, or *dust_schme=3* for S11 in the namelist.input configuration file."
* * *

---

## Author Comment (AC2) · 1 Nov 2018

**Reviewer 1**
**The manuscript provides a full documentation of the AFWA dust emission scheme in WRF-Chem, and the difference between dust emission produced by the three available dust emission schemes are also discussed. The manuscript is well-written and clearly presented.**

We thank the Reviewer for considering our manuscript and his/her positive comments.

**Suggestion on the manuscript.**

**The authors should apply more observational data to show the difference be-
tween the results produced by the three dust emission schemes? Such as the
daily AOD map from MODIS, or the surface AOD observations from AERONET
program? It is hard to conclude with only the transects of CALIPSO extinction
coefficient.**

We thank the Reviewer for the suggestion. Figure 1 (new Fig. 8 in the manuscript)
provides a comparison of simulated 8-hour average 550nm AOD (centered at 25 Jan
2010 10:00 UTC) to the 1km-resolution MCD19A2 MODIS daily AOD product from 25
Jan 2010 (provided by the NASA Land Processes Distributed Active Archive Center (LP
DAAC), USGS/Earth Resources Observation and Science (EROS) Center, Sioux Falls,
South Dakota [https://lpdaac.usgs.gov/data_access/data_pool]). The effect of clouds
on the MODIS AOD retrieval is evident as much of the AOD in the image is masked out.
In areas that are cloud free, we find the results are similar to the analysis presented
in our CALIPSO comparison discussion. A regional peak in AOD is observed near the
border of Iraq and Saudi Arabia. The general patterns of average AOD simulated for
the same time period by the GOCART-WRF scheme are broadly consistent with the
MODIS AOD product. Simulated AFWA scheme AOD is too strong over eastern Iraq
but captures the extent of the plume across the southern half of Iraq towards Kuwait.
There is less agreement with the UoC scheme, which produces several localized, high
AOD values over Syria, Jordan, and western Iraq instead of the broader AOD patterns
generated by the other two schemes.

**MODIS (MCD19A2) AOD**

a)

**GOCART-WRF**

c)

**AFWA**

b)

**UoC**

d)

**Fig. 1.** The (a) MCD19A2 MODIS AOD product for 25 Jan 2010 and simulated 8-hour average 550nm AOD centered at 25 Jan 2010 10:00 UTC for (b) GOCART-WRF, (c) AFWA, and (d) UoC.

---

## Author Response (AR1)

**Response to reviewers:**

**The AFWA Dust Emissions Scheme for the GOCART Aerosol Model in WRF-Chem**

**LeGrand et al.**

**Executive Editor comment on gmd-2018-169**
*Thus, at least add to the title the version number of the WRF version which includes the exact dust emission schemes discussed here.*

We have changed the title of the manuscript to include a version identifier (v3.8.1) as requested by GMD executive editor Astrid Kerkweg.

*Regarding the Code availability, please ensure that the exact version described here is available to the reader. Additional, add the information how to access the code.*

We have updated the code availability section to include a more direct link, changed the WRF-Chem version number from the most recent release to the specific version used in this study, and listed key configuration file settings required to run the WRF-Chem model with the dust emission schemes.

**Reviewer 1**
*The manuscript provides a full documentation of the AFWA dust emission scheme in WRF-Chem, and the difference between dust emission produced by the three available dust emission schemes are also discussed. The manuscript is well-written and clearly presented.*

We thank the Reviewer for considering our manuscript and his/her positive comments.

*Suggestion on the manuscript.*
*The authors should apply more observational data to show the difference between the results produced by the three dust emission schemes? Such as the daily AOD map from MODIS, or the surface AOD observations from AERONET program? It is hard to conclude with only the transects of CALIPSO extinction coefficient.*

We thank the Reviewer for the suggestion and have added a comparison of simulated 8-hour average AOD to the daily MODIS AOD product for 25 January 2010. We find the results are similar to the analysis presented in the CALIPSO discussion.

We have updated the manuscript with the following.
P24 L7 – "We also use the 1km-resolution MODIS MCD19A2 daily AOD product (Lyapustin and Wang 2018) provided by the NASA Land Processes Distributed Active Archive Center (LP DAAC; https://lpdaac.usgs.gov/data_access/data_pool) to quantitatively evaluate the simulated AOD." The header for section 5.1 was also updated to include MODIS AOD.

P27 L10 - "Figure 8 compares simulated 8-hour average 550nm AOD centered at 1000 UTC 25 January 2010 to the MCD19A2 MODIS AOD product from 25 January 2010. The effect of

clouds on the MODIS AOD retrieval is evident, as much of the AOD in the image is masked out. A regional peak in AOD is observed near the border of Iraq and Saudi Arabia. The general patterns of average AOD simulated for the same time period by the GOCART-WRF scheme are broadly consistent with the MODIS AOD product in the southern part of Iraq and over the Persian Gulf. An area of high AOD in northern Iraq is challenging to compare to observations due to a lack of data in much of that region. Simulated AFWA scheme AOD is too strong over eastern Iraq, and also appears to be placed west of the observed plume, perhaps due to a mismatch in timing of emission and therefore less downwind transport, but still captures the extent of the plume across the southern half of Iraq towards Kuwait. Again, high AOD in northern Iraq is difficult to assess. There is a mismatch between the high AOD modeled by the AFWA scheme in northwestern Iraq and observations, but a lack of data just east of the simulated plume location prohibits assessing whether there is simply a small temporal mismatch. There is less agreement with the UoC scheme, which produces several localized, high AOD values over Syria, Jordan, and western Iraq instead of the broader AOD patterns generated by the other two schemes."

**Reviewer 2**

*The authors describe and compare the current dust emission options in WRF-Chem and discuss similarities and differences between the different options. The objective of the paper is to document the AFWA-dust emission module in WRF, but strong emphasis is given also on the GOCART and UoC dust modules with the goal to compare the implementations and document so-far undocumented aspects. While this is useful it does not seem to have happened with interaction/consultation of the persons responsible for the implementations, which is – at least – surprising and which might have helped to clarify certain aspects.*
*The paper is overall well written and organized. However, there are several shortcomings/incorrect statements, in particular regarding the description of the UoC implementation. I also see some problems regarding the terminology and code versions used for the simulations. I recommend revision of the manuscript, considering the following comments:*

We would like to thank the Reviewer for carefully reading our manuscript and for his/her very helpful comments. The detailed and thorough review they provide is greatly appreciated and has caught several errors. We greatly appreciate the Reviewer's help in bringing a complete and accurate documentation of these models to publication and have addressed each of the comments they raise below.

*GOCART-WRF Implementation:*
*The authors discuss the change of an expression for the saltation threshold in the GOCART-WRF implementation from one for wind velocity to one for friction velocity. It is important to note here that both equations for threshold velocity (Eqs. 2 and 5) were originally expressions for threshold friction velocity, only the coefficient A in Eq. 2 was adapted, supposedly to mimic a wind speed rather than friction velocity. The deficits discussed in section 3.1.2 could therefore be easily overcome by either doing a similar empirical adjustment or by using one of the stability functions to convert between u\* and u readily available from the surface layer physics in WRF. The authors further discuss that the use of such a threshold friction velocity would be "physically invalid" (P9 L16), because it is designed to represent the initiation of saltation (P8 L26) while*

*saltation is not explicitly represented in the GOCART-WRF scheme. This argument does not hold, because the merging of saltation and dust emission to one empirical relationship in the parameterization does not contradict the assumption that dust emission is initiated by saltation. This is also stated by the authors themselves (P5 - L15-19): "The impacts of saltation bombardment processes on mobilization are not necessarily omitted - rather they are internalized in the relationship between wind speed and emissions". For this reason, I suggest still to highlight the issue of comparing u\* with u in the current implementation also mentioning that a correction like it was done before could easily be added, but to remove the discussion about the unphysical use of the equation (in an empirical parameterization) at the end of Section 3.1.2, the purpose of which seems to be mainly to motivate the introduction of the AFWA module. This is unnecessary. The formulations in this motivating paragraph, i.e. P9 L14-L22, to me also seem to be too strong statements in terms of the novelty of the implementation keeping in mind that it is not a new emission parameterization, but the incorporation of existing and well-known parameterizations in WRF. Apart of that, I recommend to add references to Eqs. (2) [Bagnold, 1941; Ginoux et al. (2001)].*

In response to this comment, we have removed the statement about the MB95 function being used in a non-physical manner and better clarified the difference between AFWA and GOCART-WRF, namely AFWA captures the two-step saltation bombardment-dust emission process more explicitly. Regarding the Reviewer's note that we over-represented the novelty of the AFWA implementation, we did not intend to imply that the AFWA functions were novel but see how the words could easily be interpreted that way. We changed the wording to clarify that replacing "new parameterization" to more clearly convey that, relative to the simplicity of GOCART-WRF's combined saltation bombardment-dust emission function, the AFWA scheme uses an *additional* function – making it a two-step process. We also added the suggested citations to Eq. (2).

*- P6 L15 The authors state that the impact of a soil moisture correction factor > 1 is small, because soils moisture does not normally assume such small values "in most numerical weather models". It would be more relevant here to discuss this in the framework of WRF which does seem to allow for such small values (P28 L23-24).*

The Reviewer is correct that this limitation does not apply in WRF-Chem (or WRF). We have removed the statement about "most numerical weather models" and agree that it is irrelevant here.

*- P8 L7-9 The mismatch between predicted and observed threshold friction velocities for small particles in the Bagnold-parameterization is well-known and dates back to the mid/late 20th century. Iversen and White (1982) provided the next well-referenced parameterization for u\*t including a minimium of u\*t for particles of about 100 micrometers in diameter (Iversen and White, 1982 is also the basis for the MB95 expression used in the AFWA implementation), followed by Shao and Lu (2000), who put the expression on pure physics-based footing. Reference to a modeling study from 2003 does therefore not seem appropriate here.*

Our intent here was to acknowledge other authors for previously identifying the small particle lofting threshold issue in the original GOCART dust emission scheme prior to this work. After

revisiting this section, we agree that our original phrasing was confusing and have changed P8 L7-9 to "Note that at a given soil moisture content, threshold wind velocity in this formulation is always greater for larger particle diameters, a known issue with the GOCART dust emission scheme (e.g., Colarco et al., 2003a; Ginoux et al., 2004)." We also updated the references listed in the sentence immediately following to include the citations recommended by the Reviewer: "Well-established experimental observations instead show particles below ~60 μm in size exhibit higher threshold wind speeds with decreasing diameter due to the increasingly dominant influence of cohesive effects on smaller particle binding (e.g., Bagnold, 1941; Iversen and White, 1982; Alfaro et al. 1998)."

*- P8 L18-19 It is explained here that the coefficient in Equation 15 (0.129) is given as 0.0013 in the model due to rounding and unit conversion. However, checking the source code, I see a factor of 0.13 (L. 273 in module_gocart_dust.F and L.511 in module_gocart_dust_afwa.F, WRF-Chem V4.0). Please clarify.*

We confirmed our original value. It is possible the reviewer missed the scientific notation. The coefficient used in both modules in the model is 0.13 x 1.0D-2, or 0.0013.

*AFWA implementation:*
*- P9 L26 The MB95 parameterization represents saltation bombardment only.*

The sentence was clarified to indicate that the two-part saltation bombardment- dust emission description applied to the AFWA scheme rather than the MB95 parameterization.

*- Repetition of Eq. (5) seems unnecessary here.*

We found it was helpful during our internal review process to repeat key equations for in-depth comparison discussions to improve readability, especially given the length of the paper.

*- P9 L12 - See previous comment on the factor 0.0013*

See above. We confirmed that the 0.0013 factor is correct, no change is required.

*- Please add reference to Eq. (10)*

Done.  Equation 10 is calculated following Kawamura (1951).

*UoC implementation:*
*- P14 L7 The namelist variable is called dust_schme and not dust_scheme.*

Corrected. Thank you for catching that.

*- P14 L12 ["Both schemes simulate the physics of dust emission"] This is not correct. While the Shao schemes used in the UoC module are physics-based parameterizations, the AFWA module makes use of the Marticorena and Bergametti parameterization, which is semi-empirical. See*

*also my later comments on "physics-based schemes" and the technical term "schemes" under "Terminology"*

We agree with the Reviewer's comment about the UoC scheme being more physics-based than AFWA scheme. Our goal with this section was to imply that the UoC scheme is more like the AFWA scheme than the GOCART-WRF scheme in that it includes separate calculations for the horizontal saltation flux and the vertical dust emission flux. The second sentence of the paragraph beginning on P14 L12 has been changed to the following: "Both schemes simulate dust emission by first calculating a threshold friction velocity for particle saltation, then using that threshold friction velocity to determine saltation flux, and finally calculating emissions of dust particles caused by saltation processes (e.g., bombardment), capturing the general process of dust emission more fully than the GOCART-WRF scheme."

*- P14 L15 Which dust emission bins are referred to here, the bins to calculate the emissions or the bins passed on to the WRF transport routines? The former are not the same between the UoC and AFWA modules and the latter are consistent with the GOCART-WRF and AFWA implementations only from WRF V3.8.1. Before that the UoC implementation was using different bins (see Flaounas et al., 2017)*

We thank the Reviewer for catching this discrepancy. All three schemes use the same five dust bins to pass emitted dust to the WRF transport routines (0.2-2, 2-3.6, 3.6-6, 6-12,12-20 μm) from WRF-Chem v3.8—4.0.1. Note the effective diameter sizes for bin 2 and bin 4 are slightly different than those reported in Flaounas et al. (2017)). The default emitted dust size bin settings for the GOCART-WRF and AFWA schemes have been consistent since their original release to the user community. In WRF-Chem v3.6.1—3.7.1, the UoC scheme used four size bins (<2.5, 2.5-5, 5-10, 10-20 μm) to pass emitted dust to the WRF transport routines. Flaounas et al. note this change in implementation in their study using WRF-Chem v3.6.1; however, the code change does not appear to have been added to the community baseline until v3.8. We have removed the P14 L15 statement "Both schemes also use the same size-resolved dust emission bins" from the manuscript and added the following to P21 L10 Point 6 - "We also note a change in the number of dust size bins used to pass emitted dust from the UoC scheme to the WRF-Chem transport routines. Four size bins with diameter ranges of <2.5, 2.5-5, 5-10, and 10-20 μm are used in v3.6.1—3.7.1. These size bins were reconfigured to match the five bins used in the GOCART-WRF and AFWA schemes (0.2-2, 2-3.6, 3.6-6, 6-12,12-20 μm), starting with v3.8."

*- Note that while Eq. (17) might give similar output like Eq. (5), it is not empirical.*

Agreed. This is an important distinction between the two approaches. We updated the text from P14 L18-20 to better emphasize this point: "The calculation of the threshold friction velocity for initiation of particle saltation used by the UoC scheme is physically-based and of significantly different form, compared to the semi-empirical MB95 function used in the AFWA scheme, but has similar output in terms of calculated threshold friction velocity $u_{*t}$ under a given set of forcing conditions. Equation (5) and Eq. (17) serve this equivalent function for the AFWA and UoC schemes, respectively…"

*- P14 L25 The value of 1.65 x 10ˆ-4 kg sˆ-2 is documented in Darmenova et al. (2009)*

We appreciate the Reviewer pointing us to the Darmenova et al. (2009) reference. We feel it will be helpful to the community to keep our discussion about the discrepancy between the WRF-Chem implementation and the original scheme description to help users follow the evolution of the code over time. We've updated the discussion to reflect that the value of $\gamma_c$ used for UoC has also been adopted by Zhao et al. (2006), Park et al. (2007), and Darmenova et al. (2009):

"As we will note in documenting code discrepancies below, $\gamma_c$ is set to 1.65 x 10$^{-4}$ kg s$^{-2}$ in the code (a value of $\gamma_c$ also adopted by Zhao et al. (2006), Park et al. (2007), and Darmenova et al. (2009)), while it is specified as 3.0 x 10$^{-4}$ kg s$^{-2}$ in Shao and Lu (2000)."

- P15 L18 I strongly recommend not to merge coefficients here, as this can give an equation a different appearance. Please list all coefficients separately for consistency with the original references.

It seems possible that the reviewer is looking at a different version of the equation but coefficients are not merged relative to Shao et al., 2011. The equation listed matches quite closely with Shao et al., 2011 Eq. 19. We have added a citation to clarify this as the source.

*- P15 L7-8 The UoC implementation uses the vegetation fraction provided by the WRF model. This can easily and should be updated for case studies to obtain more accurate results. The specific vegetation product used is therefore not a feature of the UoC dust emission module, but of the parent WRF model.*

We fully agree with the Reviewer's comment about the WRF-supplied vegetation fraction settings. It's an issue that also affects other terrain attributes important to dust emission processes (e.g., roughness length, soil type, soil mass fraction, land use/vegetation type, etc.). As such, we've update P15 L7-8 to better reflect the source of the input parameter:

"Vegetation fraction ($c_f$) is set using the *greenfract* variable from the parent WRF-Chem model, which as of this writing, is determined from the MODIS Fraction of Photosynthetically Active Radiation (FPAR) absorbed by green vegetation monthly climatological values in the default WRF-Chem configuration."

However, we're hesitant to suggest that a user should automatically alter terrain input datasets to obtain better results without consideration for how other aspects of the WRF-Chem model (e.g., land surface and boundary layer schemes) will respond.

*- P16 L2-3 The statement here is unclear and misleading. Supply-limited saltation is not accounted for in either of the implementations in WRF. While the EROD function is meant to represent the availability of erodible sediment, it does by no means account for supply limitation in its physical meaning within the saltation process. Rather, it represents the "most probable locations of sediment" (Ginoux et al., 2001).*

There was unintended meaning in what we wrote, and we appreciate the Reviewer catching this. We modified the text to clarify that the EROD function is not accounting for supply limitation by removing references to erodibility.

*- P16 L7-9 This sentence is not clear to me.*

We have changed P16 L7-9 to "This is in contrast to the AFWA scheme, which handles all soil particles according to a single fundamental particle size distribution (see Eqs. (11) and (12). Saltation in each bin in AFWA is also affected by the relative surficial area coverage of each particle class rather than the bulk particle fraction." to help clarify.

*- P16 L10 the variable dpsds is not calculated using Eq. (22). Eq. (22) gives the probability density function for airborne sediment particle-size distribution p_s(d) ("psds" in the code) (e.g. S11). Please modify Eq. (22) accordingly for consistency with S11. "dpsds" is the probability for each bin and follows according to the definition of probability density functions. There is therefore no need to introduce such an internal variable here.*

We thank the Reviewer for pointing out the terminology and symbology error. We've changed the sentence starting on P16 L10 to "The term capturing the probability density function for airborne sediment particle-size distribution is calculated according to Eq. (22) (equivalent to Eq. (8) in S11):" and updated the symbology in Eq. (22), (21), and the symbol table in the appendix.

*- P16 L14 d is diameter, not bin.*

Corrected.

*- P16 L15-16 ["Limitations..."] This seems to be a general statement and not specific to the UoC implementation.*

Agreed. We've removed the statement from the manuscript.

*- P16 L19-20 ["prior to correction for soil moisture and ground cover"] This is not correct; the corrections are applied first.*

We thank the Reviewer for pointing this out. We've checked the code and agree. P16 L19-20 has been changed to "… $u_{*t}$ is the threshold friction velocity from Eq. (17) with the corrections for soil moisture and roughness applied."

*- P17 L7 ["other tuning parameters"] While soil characteristics like the ones mentioned can be used to tune a model, they are not per se tuning parameters, but have a physical meaning.*

We thank the Reviewer for the terminology suggestion. P17 L7 phrasing has been changed to "other soil attributes."

*- Eq. (25) I do not understand how the authors derived this equation. It is inconsistent with the one implemented in the UoC-S01 module. See also my comment further down on Section 3.3.2, Point 6. Apart of that, it needs to be Q(d_s) rather than q(d_s).*

Thank you for finding this error. We revisited the code and our equation comparisons. The Reviewer is correct. Our Eq. (25) does not match Lu and Shao (1999) and or the vhlys function in the UoC code. The Reviewer is also correct in that Eq. (8) in Lu and Shao 99 and Eq. (36) in S01 are identical. Eq. (25) has been corrected in the manuscript with the following

$$\Omega = d \left[ \frac{U_p^2}{\beta_v^2} \left( \sin 2\alpha_i - 4 \sin^2 \alpha_i \right) + \frac{7.5\pi}{d} \left( \frac{U_p \sin \alpha_i}{\beta_v} \right)^3 \right], \tag{25}$$

and the discussion point 6 in Section 3.3.2 has been removed from the manuscript accordingly. We have corrected Eq. (24) to include Q(d_s) rather than q(d_s).

*- Eq. (27) Q(d_s) rather than q(d_s)*

Corrected.

*- P18 L16 The authors discuss here about a vegetation correction applied on both saltation and dust emission flux in the model and speculate that this correction "may be in error". The correction effectively reduces the surface area from which (a) sand particles and (b) dust particles can be emitted. Considering emission as a two-part process, application of the correction twice, i.e. for Q and F separately, is therefore plausible.*

The Reviewer makes an excellent point! We've incorporated this into the manuscript starting on P18 L16:

"In S01 and S04, the size-resolved dust emission is calculated by integrating dust emissions of each dust bin over all saltation bins. During this step, an additional factor of $1-c_f$ is applied.

$$F(j) = \left( 1 - c_f \right) \sum_{i=1}^{bins=100} F(i,j) \tag{30}$$

This factor does not appear in the papers that document these schemes (S01, S04, S11) and may be in error; however, since the correction effectively reduces the surface area from which both sand particles and dust particles can be emitted, application of the correction twice (i.e., once for saltation and once for dust emission) may be physically valid."

*- P19 L6 The authors claim that "measurements of these soil characteristics are generally unavailable", referring to the use of soil particle-size distributions. This is surprising given that a complete set of parameters representing particle-size-distributions for the 12 USDA soil-texture classes is provided with the implementation. Availability is therefore not an issue and can be considered similar to that of other "difficult-to-obtain" soil-related parameters, e.g. porosity or clay fraction as used in the AFWA implementation.*

We do agree that spatially-varying soil attribute datasets could easily be added to the WRF-Chem framework, but the fully-disturbed and minimally-disturbed soil particle size distribution and the soil plastic pressure are not widely *measured* variables. Though a data layer is available, these data have a limited measurement foundation. Something like clay fraction is much more commonly measured.

*- P19 L10-18 The description of how the soil particle-size distributions are obtained is not clear. The use of the FAO soil map is again, like vegetation cover, that provided by the WRF modeling framework and should not be considered as a feature of the implementation. The term "soil modes" is also misleading in the context of probability density functions, for which a "mode" has a statistical meaning. The soil parameters available in the UoC implementation are assigned to the 12 USDA soil texture classes for each of which particle-size distributions can be computed. Further, the particle-size distributions are calculated in the subroutine psd_create and not in the subroutine h_c. The latter determines the moisture correction of the threshold friction velocity. However, I believe that the names of individual subroutines should not be discussed here*

Discussion of subroutines by name is removed as requested. The clay and sand fractions referenced here were not originally part of the WRF framework. These two soils datasets were provided to us by the NASA LIS community and submitted with the AFWA scheme code to the WRF-Chem repository. To the best of our knowledge, these datasets are not used outside of the AFWA and UoC dust emission schemes.

*- In the original paper S04, c_y varies from 1 x 10ˆ-5 to 3 x 10ˆ-4. Note that exponential notation (1 x 10ˆ-5 rather than 1e-5) is preferable.*

Corrected.

*- Sec. 3.3.2, Point 2 - documented in Darmenova et al. (2009), see comment above*

Please see response to comment above. We would like to retain the text as is with the following addition so users can follow the evolution of the code: "Our mention of this discrepancy, however, is only to bring awareness to the model user. As discussed by Darmenova et al. (2009), $\gamma_c$ can be thought of as a tuning parameter for adjusting the onset and magnitude of modeled dust emission."

*- Sec. 3.3.2, Point 3 - The roughness correction represents drag partition, while the application of (1-cf) correct for the area covered by vegetation. The factor is discussed in Darmenova et al. (2009).*

Please see response to comment above. Again, we would like to retain the text with the following addition so users can follow the evolution of the code: "This discrepancy between the code and literature, however, does not necessarily imply the WRF-Chem implementation is physically invalid since the presence of vegetation can affect both saltation and dust emission processes."

We changed the following text to better differentiate between the roughness correction factor and the vegetation coverage correction factor in the UoC overview:

P15 L2-3 - "In the UoC scheme, an additional correction factor, titled the roughness correction (also commonly referred to as the drag partition correction), is applied to the threshold friction velocity to account for terrain attributes that absorb wind momentum or shelter exposed soils." Section 3.4 Point 4 on P22 L10-11: "The UoC scheme incorporates a second correction factor in the calculation of threshold friction velocity for nonerodible roughness elements (i.e., a drag partition correction), which is determined from the vegetation coverage layer."

*- Sec. 3.3.2, Point 4 - The use of the Kawamura/White saltation flux equation is documented in Shao et al. (2011), in which also the Shao (2004) scheme is used.*

We agree that the Kawamura/White saltation flux equation is documented in Shao et al. (2011). However, we also note that in Shao 2001 and Shao 2004, the saltation flux equation from Owen (1964) is described and referred to, which is slightly different the Kawamura/White. We also note that in the code (module_qf03.F), the soil moisture and roughness corrected saltation flux calculated using the Kawamura/White equation is used in all three (Shao 2001, 2004, and 2011) dust emission schemes. Our purpose here is to point out that the saltation flux equation described in Shao 2001, and referred to in Shao 2004, is different than the saltation flux equation implemented in the Shao 2001, and Shao 2004 schemes in WRF-Chem. The point appears valid, and so we have left the text from point 4 as it is currently written.

*- Sec. 3.3.2, Point 5 - See earlier comment on "soil modes"*

Corrected.

*- Sec. 3.3.2, Point 6 - This point is also incorrect. First, Eq. (25) is not the one implemented in the model. In the relevant subroutine (vhlys), it is stated clearly that the subroutine computes Eq. (8) from Lu and Shao (1999). Comparing the implementation with Eq. (8) in the original paper shows that the two are in perfect agreement. The supposed difference of a factor of 1/d mentioned by the authors disappears understanding that Eq. (8) in Lu and Shao (1999), gives V/b rather than V and that b is approximately equal to d as explained in Shao (2001). The reason why the Equation from Lu and Shao (1999) is implemented here is likely the fact that the new Equation in Shao (2001) is more complicated and subject to further testing as is discussed at length in Shao (2001). Second, Eq. (36) [also Eq. 36 in Shao, 2001] is also in perfect agreement with Eq. (8) in Lu and Shao (1999), which can easily be show using mathematical conversions and inserting beta, while the Equation given by the authors (their Eq. (25)) is incorrect.*

Please see response to comment above. The Reviewer is correct. We have removed this part from the manuscript.

*- P21 L21 The Shao schemes available in the UoC module do not include aerodynamic (dust) entrainment. In Shao et al. (2001), Section 5 it is stated: "Here we are mainly concerned with the latter case" referring to saltation-based dust emission*

We thank the Reviewer for the comment and have removed the aerodynamic entrainment statement from P21 L21.

*- P21 L27 Eq. (7) in Shao (2004) does not represent sigma_p. Eq. (7) describes gamma (cf. Eq. (23) in the present paper).*

The sigma_p parameter is defined in an un-numbered equation immediately below Eq. (7) in S04. We have changed P21 L27 to "captured in sigma_p, as defined by S04."

*Test case and comparison:*
*- P22 L13 The references given here belong to WRF-Chem, not to the dust emission schemes. I suggest moving them to an earlier position.*

Done.

*- If the UoC saltation flux bug fix was released in January 2018, this was well before submission of the manuscript. The version used for evaluation in this paper should therefore be the one with the bug corrected. There is no point in using a version that is known to be wrong and that is outdated. If the authors wish to test the effect of this bug fix on the results, they can do so in an appendix, but the version in the main text should be the version "as is", i.e. including the bug correction.*

Our previous statement that a bug-fix had been released on 9 January 2018 was incorrect. An announcement and recommended correction had been sent to a select group of WRF-Chem model developers; however, a corrected version of the UoC code was not widely disseminated until the public release of WRF-Chem v4.0 on 8 June 2018, about a month before we submitted this manuscript to GMD for consideration.

This paper was written using model version 3.8.1 and begun well before January 2018. The policy of GMD is to demand papers be written on a particular, broadly-released version of the model, in order to capture a model at a point in time – not necessarily the most recent release. Though a recommended bug fix was announced in January 2018, it is not present in the current publicly available release of model version 3.8.1, and therefore it is not appropriate for us to include the corrected version in the main text (we have also not used the corrected AFWA scheme to produce results used in the main text). We are also wary of the idea of back-correcting model versions, as this can create great confusion in comparing results that a casual user feels were from the same model version.

Taking the concept, however, we have added a brief analysis to the effects of the bug-fix in an appendix.

*- In Section 3.2, an implementation error is mentioned for the AFWA implementation. It is not clear whether the version used in the comparison is the one with or without the error correction. The same as mentioned in the previous comment for the UoC scheme applies here, too, with the only difference that the correction for the AFWA scheme does not seem to be included in the current release, but will be in a future version.*

We agree with the Reviewer and have removed all discussion of AFWA scheme alterations from the main body of the text. Table 1 from our original submission has been replaced with Table 4

(the nine saltation bins and their associated attributes as currently implemented in WRF-Chem). The 10-bin saltation configuration originally presented in Table 1 has now been moved to the appendix, and we've added a brief discussion of how the change affects simulated AOD.

The following text has been added to a new appendix to provide readers with a brief overview of the effects of the UoC bug-fix and the alternate AFWA saltation bin configuration on WRF-Chem simulated AOD:

"The results and discussion presented in our study explore use of the three currently availableWRF-Chemdust emission schemes as they are presented in version 3.8.1; however, as highlighted in the text, there are some relatively easy to correct errors in the AFWA and UoC code that are worth examining further. Here, we assess the effects of the UoC saltation function order of operations error described in section 3.3.2 (i.e., Eqs. (34) and (35)) and use of an alternate configuration for the AFWA scheme saltation bins by rerunning our simulation with bug-fixes applied for comparison.

For the UoC scheme, we correct the order of operations error in the UoC saltation flux calculation (i.e., Eqs. (34) and (35)). While this error was corrected in WRF-Chem version 4.0 (released June 2018), the bug remains in all previously released versions of WRF-Chem, including version 3.8.1. For the AFWA scheme, we reran our simulation using an alternate saltation bin configuration described in Table (A1) that better aligns with the mass distributions recommended by Tegen and Fung (1994). These bin configuration changes were implemented in the existing version 3.8.1 AFWA code by altering the settings for the *ngsalt*, *reff_salt*, *den_salt*, *spoint*, and *frac_salt* parameters in the *module_data_gocart_dust.F* file according to Table A1.

Simulated 8-hour mean AODs (centered on 25 January 2010 1000 UTC) from the original and altered UoC and AFWA version 3.8.1 codes were used to illustrate the effects of these changes. Figure A1 shows the calculated difference in 8-hour mean AOD between the corrected and uncorrected versions of each scheme. The UoC scheme correction has little effect on the spatial extent of the dust plume but essentially doubles the AOD magnitude in regions where dust is present. Similarly, use of the alternate saltation bins in the AFWA scheme has a relatively negligible effect on the location and extent of the simulated dust plume. However, in contrast to the UoC correction, the AFWA AOD differences are smaller and of mixed sign.

Based on these results, we recommend that model users consider the impact of the UoC saltation flux error when assessing published results from studies performed using the UoC scheme prior to the release of WRF-Chem version 4.0. The effects of the alternate saltation bin configuration on overall AFWA scheme performance are less clear. Optimal settings for the saltation arrays may be region dependent. Further analyses beyond the scope of this paper are still needed."

*- P23 L20-21 ["The atmospheric dust observed..."] Please add reference, e.g. a figure, or give additional explanation.*

Our evidence for this statement is based on qualitative assessment of the MODIS imagery that appears to show narrow plumes of dust originating in this region (see: https://earthobservatory.nasa.gov/images/42450/dust-over-iraq) and available surface METAR

observations in the region. We have clarified this statement to directly document the available information: "The atmospheric dust plumes observed by satellite remote sensing platforms during this event appeared to originated largely in Western Iraq and Syria qualitatively indicating a large, possibly dominant, role for dust emission from this region during the event."

*- P24 L25-26 It is sufficient to give the color coding in the figure caption.*

Removed the figure color description from text.

*- P25 L28, P27 19 I suggest adding one or two more references for the "spurious dust lofting" in the GOCART-WRF implementation if available, keeping in mind that - if it depends on u\*t vs. ut - this could be relatively easily fixed.*

Published references describing the spurious lofting model behavior of GOCART-WRF are limited. US Air Force technical reports detailing model performance exist (e.g., Jones 2012), but these reports are not cleared for public distribution. Furthermore, negative outcome model studies without a replacement recommendation rarely make it into publication.

The motivation to find a replacement for the GOCART-WRF dust emission was largely driven by anecdotal reports/community feedback on GOCART-WRF model performance. Four of the participating authors on this paper (LeGrand, Creighton, Cetola, and Peckham) have extensive experience supporting operational weather forecasting centers that used the GOCART-WRF model and regularly received feedback on model behavior from operational weather squadrons and staff weather officers in southwest Asia. Dr. Peckham also served a key role on the primary WRF-Chem development team and frequently received model troubleshooting/support requests sent through the WRF helpdesk regarding unrealistic dust emissions produced using GOCART-WRF code.

*- P26 L9-10 The larger spatial extent in the results of the GOCART-WRF scheme are visible most of the time in Fig. 5, but not at 10 UTC on 25 Jan for which the MODIS data is shown in Fig. 4.*

At 1000 UTC on 25 January 2010 there is an overly large region of the domain covered by dust in the GOCART-WRF scheme that extends well beyond the region where dust was actually observed via satellite. For example, the moderate-to-high values of simulated AOD over Azerbaijan and Caspian Sea as well as the plume over the Black Sea and Russia. While there are low AOD values over some of these regions in the AWFA scheme, the substantial dust concentrations are much more confined to the region where the dust event is observed.

*- P27 L24 (and relevant subsequent passages) The binary use of the EROD function cannot cause a reduced area of active dust emission in the UoC parameterization: dust emission is possible wherever EROD > 0, i.e. wherever dust emission is possible in the AFWA implementation.*

The reviewer is correct. This disproven hypothesis is now removed from the discussion.

*- P27 L29 The version using the bug fix should be used here - see earlier comment.*

Please see earlier comments regarding our use of WRF-Chem v3.8.1.

*- P28 L5 Is the threshold friction velocity meant with "soil threshold parameter"? In that case it would depend on particle size and not be a single value.*

We agree with the Reviewer. Our intent here was to walk the reader through the various components of the lofting threshold equation, which may not have been clear in our presentation of the dry lofting threshold on a 2-dimensional map. We changed the text starting on P28 L3 to the following to help clarify:

"We begin our analysis by calculating dry soil threshold friction velocity required for initiating particle mobilization for each of the three dust emission schemes. The dry soil threshold parameter for these schemes only varies as a function of particle size (i.e., it does not vary spatially); however, we provide results in mapped display (Fig. 11, column 1) for ease of discussion with respect to the soil moisture and roughness correction factors. Resultant dry soil thresholds for given particle sizes are shaded everywhere the dust source function is nonzero.

Direct comparison between the GOCART-WRF scheme and the other two schemes is not possible since the GOCART-WRF scheme only considers dust-sized particles, but for completeness we determine the dry soil threshold velocity for a grain diameter of 16 μm (the effective diameter of the largest dust bin) to be equal to 0.48 m s$^{-1}$ using the GOCART-WRF implementation of Eq. (5). The AFWA and UoC schemes determine the dry soil threshold friction velocity based on Eq. (5) and (17), respectively. Though the calculations are different, we note that the resultant threshold for a 60 μm particle (i.e., a relatively small, easy to mobilize sand-sized particle (e.g., Bagnold, 1941)) is 0.24 m s$^{-1}$ in both the UoC and AFWA schemes (as shown in Fig. 11, column 1). We therefore conclude that minor differences in these threshold friction velocities are not a major cause of differences in AFWA and UoC dust emissions."

*- P28 L16 The coefficients used in the soil moisture correction are not only different due to different units. Different sets of coefficients are also used for each of the 12 soil texture classes (Klose et al., 2014; based on Shao and Jung, 2000, unpublished manuscript)*

We thank the Reviewer for describing this reference. P28 L14-16 is changed to "The general equation for calculating this correction in AFWA and UoC schemes is identical (Fécan et al., 1999) but we see slightly different output, presumably due to differences in coefficients assumed for each soil class considered in the UoC scheme."

*- Fig. 8, If the same meteorology is used for all runs, it would be sufficient to show wind speed only once.*

We agree and have updated our figures accordingly. The top row of Fig. 8 has been removed, and we've added an additional figure for simulated 10m wind speed and friction velocity.

*- Fig. 9, All corrections - Why are there no values shown north-west of the Caspian Sea for the UoC implementation?*

We thank the Reviewer for bringing our attention to the figure issue. The contour range wasn't set high enough in the image plotting script when we generated the figure. The plot has been corrected.

*- P 29 L12-21 See previous comments on bug fix.*

Please see earlier comments regarding our use of WRF-Chem v3.8.1.

*- P29 L32/Fig. 9 Please explain why S/(rough+cf˜2) is plotted here.*

We have removed this particular plot from the discussion section and have taken a new approach for describing the influence of terrain attributes on the UoC emission fluxes. Specifically, we reorganized and modified the intermediate variable plot figures (originally Fig. 8 and 9; now Fig. 9, 10, 11, and 12) for better organization/flow of concepts and to help clarify the contribution of each intermediate parameter to simulated dust emission patterns. The new Fig. 9 shows static terrain attributes, including the source strength and vegetation fraction. Plots of threshold friction velocity, threshold friction velocity corrections, and saltation plots for a given grain size are shown in Fig. 11. The updated version of what was Fig 8 and 9 includes updated, scheme-relevant symbology and removal of the 1-c_f factor from the calculation used to generate the UoC saltation plot to better differentiate the role of the roughness correction from the vegetation correction on the spatial extent of UoC saltation and dust emission flux. The UoC 1-c_f vegetation correction factor, squared to account for the application of the multiplier in both the saltation and emission flux calculations, is now plotted in Fig. 12. Our original approach of combining the roughness correction and vegetation correction in a single plot has been removed.

*Terminology:*
*- The terms scheme, parameterization, and model are used almost interchangeably here. This is problematic, in particular in the context of the GOCART, AFWA and UoC "schemes", which in my opinion are neither scheme nor parameterization nor model, but only the implementations of existing parameterizations/schemes in a model (which would be WRF-Chem in this case). I think it is important to use consistent terminology throughout the paper.*

We thank the Reviewer for pointing out the language inconsistency and have updated the paper accordingly. GOCART-WRF, AFWA, and UoC codes are now referenced as schemes throughout the manuscript. Though we agree with the Reviewer that GOCART-WRF, AFWA, and UoC codes are technically modules of existing or modified parameterizations, our use of the term "scheme" is consistent with common usage of the phrase in the WRF-Chem community and several of the publications cited in this paper (including articles published in GMD and ACP).

*- The authors use the expression "emission mode" at several locations (e.g. P4 L3, P4 L15, P5 L20). I am not aware of any common use of this expression in the dust emission/aeolian community. I would therefore strongly recommend to abstain from this expression. Most likely it*

*is being confused with the modes of particle motion, which are, e.g., saltation, suspension, creep (Bagnold (1941), Shao (2008), Kok et al. (2012)). Please revise.*

We thank the Reviewer for the suggestion. Our intent was to introduce the reader to the three mechanisms for dust emission using terminology made popular by Shao 2008 and Shao et al. 2011. We also agree with the Reviewer that use of the term "mode" is inappropriate here and have replaced with the term "mechanism" throughout section 2.

*- P5 L16-18 The explicit separation of saltation and dust emission fluxes in a parameterization does not necessarily make it a physics-based parameterization. If the saltation flux and/or dust emission flux are represented by empirical relationships rather than basic physics, it will still be (semi-)empirical. The text should be modified accordingly.*

We have changed the sentence beginning on page 5, line 15 to read: "The scheme is relatively simple and highly empirical as compared to other dust emission schemes since its equations represent a direct…"

*Minor comments:*
*P1 L13 - particles rather than particulates*

Corrected.

*P2 L5 - Reference for GOCART model needed here, in particular the dust component that is of relevance for this paper.*

Done - Added citations for Chin et al. (2000) and Ginoux et al. 2001.

*P2 L9 - "enabling their vertical movement" is not correct here speaking of dust emissions - Please revise, e.g. ""enabling dust transport in the atmosphere"*

Done.

*P2 L11 - As the present paper is concerned with dust emission, the addition of Ginoux et al. (2001) as a reference here would be appropriate.*

Done.

*P3 L9-10 - Implementation described in Darmenova et al. (2009)*

We respectfully disagree with Reviewer on use of this reference for the UoC scheme. Darmenova et al. (2009) describes an implementation of the Shao schemes; however, the moisture correction and saltation flux are different than the UoC implementations.

*P3 L22-23 - aerodynamic lift, saltation bombardment, and particle disaggregation are not forces, but processes. The half-sentence introducing those is misleading.*

We thank the reviewer for the comment. P3 L22-23 has been changed to "Three processes are responsible for the entrainment of atmospheric dust particles: (1) aerodynamic lift, (2) saltation bombardment, and (3) particle (Shao, 2008)."

*P9 L25 saltation bombardment*

Done.

*P9 L29 "effective particle size" rather than "effective aerosol size"*
Done.

*P22 L18-19 reference to NOAA/NCEP (2000) in parentheses*

Done.

**Primary changes to the manuscript:**

- Added a comparison of simulated mean 8-hour AOD to the MODIS MCD19A2 daily AOD product.
- Addressed issues with inconsistent verbiage throughout the manuscript – particularly with respect to terms like "model", "parameterization", "scheme", and "mode".
- Corrected several errors in the UoC documentation section and added additional code/documentation mismatch information uncovered through the review process.
- Reorganized and modified the intermediate variable plot figures (originally Fig. 8 and 9; now Fig. 9, 10, 11, and 12) for better organization/flow of concepts and to help clarify the contribution of each intermediate parameter to simulated dust emission patterns. The new Fig. 9 shows static terrain attributes, including the source strength and vegetation fraction. Figure 10 shows the simulated wind speed and friction velocity (plotted separately per the Reviewer's recommendation). Plots of threshold friction velocity, threshold friction velocity corrections, and saltation plots for a given grain size are shown in Fig. 11. The updated version of what was Fig 8 and 9 includes updated, scheme-relevant symbology and removal of the 1-c_f factor from the calculation used to generate the UoC saltation plot to better differentiate the role of the roughness correction from the vegetation correction on the spatial extent of UoC saltation and dust emission flux. The UoC vegetation correction factor, squared to account for the application of the multiplier in both the saltation and emission flux calculations, is plotted in Fig. 12. Our original approach of combining the roughness correction and vegetation correction in a single plot has been removed.
- Removed all discussion of AFWA scheme alterations from the main body of the text. Table 1 from our original submission has been replaced with Table 4 (the nine saltation bins and their associated attributes as currently implemented in WRF-Chem). The 10-bin saltation configuration originally presented in Table 1 has now been moved to Appendix A, and we've added a brief discussion of how the change affects simulated AOD.
- Discussed the effects of the UoC saltation bug-fix on simulated AOD in Appendix A.
- Added a conditional to Eq. (14). Dust is only able to loft from grid cells with roughness length less than or equal to 20cm, which correspond to areas designated by the parent WRF-Chem model as grassland, sparsely vegetated, and barren land use areas. We added this to the equation to ensure complete documentation; however, this aspect of Eq. (14) has little bearing on the outcome of the case study (as shown in the new Fig. 9).
- Corrected description of the optional run time tuning parameter $c_{ustune}$ in Table 3. The $c_{ustune}$ parameter was introduced to the community (via word of mouth/email) as a means for tuning the threshold fiction velocity in the AFWA scheme. In the code, however, $c_{ustune}$ is used to adjust the friction velocity, a modification that does not affect $u_*$ values in other parts of the WRF-Chem model. This description error in our manuscript was brought to our attention by a GMDD reader via email. Box 2 of Figure 1 was also updated to reflect this change. The optional tuning parameters were not used in our simulation. Thus, this update has no bearing on our case study results or discussion.
- Corrected a missing subscript on P17 L19: $U_p = 10u_*$.
- Corrected a few minor misspellings, duplicate words, and punctuation errors.

[revised manuscript text omitted]
(d_i, d_s) = c_y [(1-\gamma) + \gamma\sigma_p] \frac{q(d_s)\,g}{mu_*^2} \frac{Q(d_s)\,g}{mu_*^2} (\rho_b \eta_{f,i} \Omega + m\eta_{c,i}), \tag{24}$$

where $c_y = 0.00001$ is a dimensionless constant, $\gamma$ is evaluated as in Eq. (21), $\eta_{f,i}$ and $\eta_{m,i}$ are, respectively, the fully- and minimally-disturbed dust fraction in bin $d_i$, $\rho_b = 1000$ kg m$^{-3}$ is the assumed bulk density of the soil, $\eta_{c,i}$ is the fraction of soil available for disaggregation ($\eta_{f,i} - \eta_{m,i}$), $\sigma_p = \frac{\eta_{f,i}}{\eta_{m,i}} = \frac{p_f(d_i)}{p_m(d_i)}$, $m$ = mass of the particle, and $g$ is the gravitational constant in m s$^{-2}$. The term $\Omega$ represents the efficiency of dust emission from bombardments or collisions and is implemented in the  scheme after Lu and Shao (1999) as

$$\Omega = \frac{mU_p^2}{2\varrho d\beta_v^2} d \left[ \frac{U_p^2}{\beta_v^2} \left( \underline{\sin 2}\sin 2\alpha_i \underline{-4\sin}-4\sin^2\alpha_i \right) + \frac{7.5\pi}{d} \left( \frac{U_p \sin(\alpha_i)}{\beta_v} \frac{U_p \sin\alpha_i}{\beta_v} \right)^{\underline{3}3}_{-} \right], \tag{25}$$

where $U_p$ is the impact velocity, $\beta_v = \sqrt{\frac{2\varrho d}{m}}$, $\varrho$ is soil plastic pressure, $\alpha_i$ is the incidence angle of the collisions, $m$ is the particle mass, and $d$ is the particle diameter.

S04 simplifies the scheme for estimating the dust emission from saltation collisions by fixing several of the free variables in Eq. (25) which were not readily available in measurements, including setting the collision angle to 15 degrees, setting $U_p = 10u_*$, and setting the particle density to 2.6 times the soil bulk density. This allows a revised form of the equation for bombardment efficiency to be derived which is particle size independent

$$\sigma_m = 12u_*^2 \frac{\rho_b}{\varrho} \left( 1 + 14u_* \sqrt{\frac{\rho_b}{\varrho}} \right), \tag{26}$$

where $u_*$ is the friction velocity, $\rho_b = 1000$ kg m$^{-3}$ is bulk soil density, and $\varrho = 30000$ N m$^{-2}$ is the soil plastic pressure. We note, in particular, the very strong role that soil plastic pressure plays in the emission through this term, and further note that the value for soil plastic pressure is set to a constant in the  WRF-Chem implementation, despite being a parameter well known to be subject to variations with soil type. Incorporating $\sigma_m$ into the dust emission flux equation and simplifying results in Eq. (27); the revised flux equation used by S04 (S04 Eq. (6))

$$F(d_i, d_s) = c_y \eta_{f,i} [(1-\gamma) + \gamma\sigma_p] \frac{q(d_s)\,g}{u_*^2} \frac{Q(d_s)\,g}{u_*^2} (1 + \sigma_m). \
[revised manuscript text omitted]